



# Role of equatorial planetary and gravity waves in the 2015–16 quasi-biennial oscillation disruption

Min-Jee Kang[1], Hye-Yeong Chun[1], and Rolando R. Garcia[2]

[1]Department of Atmospheric Sciences, Yonsei University, Seoul, South Korea
[2]National Center for Atmospheric Research, Boulder, Colorado, USA

*Correspondence to*: Hye-Yeong Chun (chunhy@yonsei.ac.kr)

**Abstract.** In February 2016, the descent of the westerly phase of the quasi-biennial oscillation (QBO) was unprecedentedly disrupted by the development of easterly winds. Previous studies have shown that extratropical Rossby waves propagating into the deep Tropics were the major cause of the 2015–16 QBO disruption. However, a large portion of the negative momentum

forcing associated with the disruption still stems from equatorial planetary and small-scale gravity waves, which calls for detailed analyses by separating each wave mode compared with climatological QBO cases. Here, the contributions of resolved equatorial planetary waves [Kelvin, Rossby, mixed-Rossby gravity (MRG), and inertia-gravity (IG) waves] and small-scale convective gravity waves (CGWs) obtained from an offline CGW parameterization to the 2015–16 QBO disruption are investigated using MERRA-2 global reanalysis data from October 2015 to February 2016. In October and November 2015,

anomalously strong negative forcing by MRG and IG waves weakened the QBO jet at 0°–5°S near 40 hPa, leading to Rossby wave breaking at the QBO jet core in the southern hemisphere. From December 2015 to January 2016, exceptionally strong Rossby waves propagating horizontally (vertically) continuously decelerated the southern (northern) flank of the jet. In February 2016, when the westward CGW momentum flux at the source level was much stronger than its climatology, CGWs began to exert considerable negative forcing at 40–50 hPa near the equator, in addition to the Rossby waves. The enhancement

of the negative wave forcing in the Tropics stems mostly from strong wave activity in the troposphere associated with increased convective activity and the strong westerlies (or weaker easterlies) in the troposphere, except that the MRG wave forcing is more likely associated with increased barotropic instability in the lower stratosphere.

## 1 Introduction

The quasi-biennial oscillation (QBO) is the dominant source of variability in the equatorial stratosphere, characterized by

alternating easterly and westerly winds with a period of around 28 months (Baldwin et al., 2001). Based on the classical theory, the QBO is generated by momentum deposition by vertically propagating equatorial planetary and gravity waves (Lindzen and Holton, 1968; Holton and Lindzen, 1972). The impact of the QBO is not limited to the tropical stratosphere; the QBO modulates the strength of tropospheric convection (Collimore et al., 2003; Liess and Geller, 2012; Lee et al., 2019), Madden–Julian Oscillation (Yoo and Son, 2016; Marshall et al., 2017), and the tropical cyclone tracks (Ho et al., 2009). In addition, the




QBO affects not only the subtropical jet and the subsequent changes in the growth and life cycle of the synoptic-to-planetary scale waves in the troposphere (Garfinkel and Hartmann, 2011), but also the intensity of the stratospheric polar vortex (Holton and Tan, 1980; Anstey and Shepherd, 2014), which is strongly tied to the surface temperature and pressure distribution (Baldwin and Dunkerton, 2001). The meridional circulation induced by the QBO also changes the transport of chemical species such as ozone (Randel and Wu, 1996), water vapor (Giorgetta and Bengtsson, 1999), methane (Patra et al., 2003), etc.

Therefore, understanding the QBO is important for improving short- and long-range forecasts due to its quasi-periodical nature and global impact (Boer and Hamilton, 2008; Scaife et al., 2014).

In February 2016, sudden development of easterly winds in the middle of the westerly phase of the QBO (40 hPa) interrupted the normal descent of the westerly phase, which was the first such occurrence since QBO observations started in 1953 (Osprey et al., 2016; Newman et al., 2016). This phenomenon is called the 2015–16 QBO disruption. None of the seasonal

forecast models predicted the QBO disruption (Osprey et al., 2016), while reproducing the QBO disruption was possible only by a JAGUAR (Japanese Atmospheric General circulation model for Upper Atmosphere Research; Watanabe and Miyahara, 2009) model initialized in January 2016 (Watanabe et al., 2018). A series of studies have shown that the major cause of the QBO disruption was strong equatorward-propagating Rossby waves from the extratropics. Those studies suggested that extratropical Rossby wave generation was enhanced due to a strong El Niño (Osprey et al., 2016; Coy et al., 2017) and low

Arctic sea-ice concentration (Hirota et al., 2018). Furthermore, anomalous westerlies in the subtropical lower stratosphere, possibly caused by El Niño and the seasonal timing (Barton and McCormack, 2017), enabled the Rossby wave flux to refract toward the equatorial stratosphere.

However, there have been several cases in which the QBO disruption did not occur despite the presence of westerly winds in the subtropical lower stratosphere and strong Rossby wave flux propagating into the equator from the Northern Hemisphere

(NH) extratropics (e.g., cases in 2010/11). This is consistent with the fact that the extratropical Rossby waves generally decelerate the edge of the QBO jet, not the jet core (O'Sullivan, 1997). Thus, it is unclear how the Rossby waves decelerate the QBO jet core during the 2015–16 QBO disruption. Regarding this question, Lin et al. (2019) have shown that strong extratropical waves in a confined longitude region could make a local critical layer for the Rossby waves to break in the middle of the QBO jet. They also argued that high-frequency waves, mainly the mixed-Rossby gravity (MRG) waves, slowed down

the background wind, which facilitates the formation of a local critical layer. This result suggests that negative momentum forcing by equatorial waves is the prerequisite for the extratropical Rossby wave breaking, which motivates the current study to investigate the role of each equatorial waves to the QBO disruption from early to later stages.

Coy et al. (2016) showed that about half of the negative momentum forcing required for the QBO disruption in February 2016 can be explained by the horizontal component of Eliassen–Palm flux (EPF) divergence (EPD), which is largely attributed

to the Rossby waves that propagate from the extratropics. They mentioned that further analyses on the vertical component of the EPD are necessary given that the vertical EPF in the tropical region significantly increased during the disruption. Barton





and McCormack (2017) also recognized the non-negligible contribution of the vertical EPD, but no systematic analysis on the equatorial wave forcing has been undertaken since then.

In the present study, we examine the contributions of equatorial planetary waves, including equatorial Kelvin, Rossby,

MRG, and inertia-gravity (IG) waves, and small-scale convective gravity waves (CGWs), to the 2015–16 QBO disruption by employing the separation method of equatorial wave modes of Kim and Chun (2015) and the offline CGW parameterization by Kang et al. (2017) using the Modern-Era Retrospective Analysis for Research and Applications version 2 (MERRA-2) reanalysis data (Gelaro et al., 2017) on native model levels. This is the first study to classify the equatorial waves in detail and investigate each wave's role in the QBO disruption. Note that if the EPD is calculated without separating each wave mode, the

importance of the westward-propagating waves might be obscured by the eastward-propagating waves, given that Kelvin wave activity was enhanced during the 2015–16 QBO disruption (Kumar et al., 2018; Li et al. 2020). The second new aspect of this study is to investigate the role of small-scale CGWs. Generally, small-scale CGWs have been recognized as an important driver of the easterly QBO (Kawatani et al., 2010; Evan et al., 2012; Kim and Chun, 2015), but this has not been examined comprehensively in terms of the QBO disruption, except by Coy et al. (2016), who analyzed the role of the parameterized

gravity wave drag (GWD) provided by MERRA-2, and by Watanabe et al. (2018), who simulated the QBO disruption with JAGUAR model resolving IG waves of horizontal wavelength longer than ~200 km. However, the parameterized GWD data provided by MERRA-2 are the combination of the orographic and non-orographic GWDs, where the non-orographic GW parameterization assumes a latitudinally dependent GW source momentum flux without considering GW sources explicitly; thus, understanding the linkage between the convective source and the wave forcing was somewhat difficult. In particular, the

QBO disruption took place during a strong El Niño phase, implying that CGW activity could be much stronger than the climatology. To overcome the simplicity of the non-orographic GW parametrization in MERRA-2 and the inevitable restriction on the horizontal resolution, in this study, we provide a realistic estimate of small-scale wave drag due to CGWs ($\lambda_h < 100$–200 km, where $\lambda_h$ is the horizontal wavelength) by using an offline, convectively coupled GW parameterization (Kang et al., 2017). The magnitude of the CGW momentum flux is constrained by observational data from the super-pressure balloons

(SPBs) in the tropical region (Jewtoukoff et al., 2013), which is the only tropical in-situ observation covering small-scale GWs ($\lambda_h < 100$ km).

The aim of this paper is to provide a comprehensive picture of the 2015–16 QBO disruption, examining the contribution of all equatorial planetary and gravity waves. We first examine the extent to which each equatorial wave contributes to the momentum budget during the QBO disruption. After determining how much wave forcing was anomalous compared to the

climatology, we investigate the possible cause of the anomalous wave forcing during the QBO disruption. Section 2 of the paper describes the data and methods used in this study. Section 3 presents the main results, including quantitative estimates of momentum forcing by equatorial waves and the sources of each wave. Section 4 provides a summary of our findings followed by concluding remarks.




## 2 Data and Methods

**2.1 Data**

In this study, we use output from MERRA-2 for 37 years (from 1980 to 2016) provided on a 0.625° longitude by 0.5° latitude regular grid at 3-h intervals. We employ native model-level data for accuracy in estimating both resolved and parameterized wave forcing, especially for the equatorial waves having relatively short vertical wavelengths (e.g., MRG waves; Richter et al., 2014). The model-top pressure is 0.01 hPa with 72 layers in total including 14 layers from 100 to 10 hPa. We

utilize zonal wind, meridional wind, temperature, geopotential height, air temperature tendency due to moist processes (DTDTMST), large-scale rainfall, and convective rainfall. In addition to MERRA-2, output from the European Centre for Medium-Range Weather Forecasts (ECMWF) interim reanalysis (ERA-I; Dee et al., 2011) with a horizontal resolution of 0.75° at 6-h intervals from 2015 to 2016 is used to examine the sensitivity of resolved wave forcing on the reanalysis datasets. The 3-hourly precipitation data from Tropical Rainfall Measuring Mission (TRMM) 3B42 version 7 (Huffman et al., 2014) for 19

years (1998–2016) with a horizontal resolution of 0.25° are also used to confirm the precipitation variability in MERRA-2.

Here, we define a westerly QBO (WQBO) phase when the zonal wind anomaly from the monthly climatology divided by its standard deviation exceeds +0.5 both at 30 hPa and 50 hPa for at least 4 months during the 6 months from October to March, the month when the QBO disruption develops. Based on this definition, 10 winters among 37 years are selected: 1980–81, 1982–83, 1985–86, 1987–88, 1990–91, 1999–2000, 2006–2007, 2008–09, 2010–11, 2013–14. The average of those 10 winters

will be referred to WQBO climatology hereafter.

### 2.2 Transformed Eulerian-mean (TEM) momentum equation

We use the transformed Eulerian-mean (TEM) zonal momentum equation in log-pressure coordinates (Andrews et al., 1987) to examine the zonal wind acceleration, resolved wave forcing, and the vertical advection:

$$\frac{\partial \overline{u}}{\partial t} = \left(f - \frac{1}{a \cos \phi} \frac{\partial}{\partial \phi}(\overline{u} \cos \phi)\right) \overline{v}^* - \overline{w}^* \frac{\partial \overline{u}}{\partial z} + \frac{1}{\rho_0 a \cos \phi} \nabla \cdot F + \overline{X}, \tag{1}$$

Here, $a$, $u$, and $\rho_0$ are the radius of the Earth, zonal wind, and air density, respectively. The overbar and prime indicate the zonal average and the perturbation from it, respectively. The EPF, expressed as $F \equiv (0, F^\phi, F^z)$, is composed of meridional $[F^\phi = \rho_0 a \cos \phi \, (-\overline{u'v'} + \overline{u}_z \overline{v'\theta'}/\overline{\theta}_z)]$ and vertical $[F^z = \rho_0 a \cos \phi \, (\hat{f} \overline{v'\theta'}/\overline{\theta}_z - \overline{u'w'})]$ components, and its divergence

(EPD) is calculated as follows:

$$\frac{1}{\rho_0 a \cos \phi} \nabla \cdot F = \frac{1}{\rho_0 a \cos \phi} \left[\frac{1}{a \cos \phi} \frac{\partial}{\partial \phi}\left(F^\phi \cos \phi\right) + \frac{\partial F^z}{\partial z}\right]. \tag{2}$$





In Eq. (1), $\hat{f}$ is the modified Coriolis parameter defined by $\hat{f} = f - 1/(a\cos\phi)\,\partial/\partial\phi\,(\bar{u}\cos\phi)$ where $f$ is the Coriolis

parameter, and $\bar{v}^*$ and $\bar{w}^*$ are the residual meridional and vertical velocities defined by $\bar{v}^* = \bar{v} - \rho_0^{-1}(\rho_0\overline{v'\theta'}/\bar{\theta}_z)_z$ and $\bar{w}^* =$

$\bar{w} + (a\cos\phi)^{-1}\left(\cos\phi\,\overline{v'\theta'}/\bar{\theta}_z\right)_\phi$, respectively. $\bar{X}$ in Eq. (1) represents forcing by processes other than EPD, including

parameterized GWD. Although MERRA-2 provides GWD, it is not coupled with the variation of the convection. Therefore,

we calculated physically based and convection-dependent GW parameterization offline (Kang et al., 2017; 2018), which will

be described in Sect. 2.4.

**2.3 Classification of the equatorial wave modes**

We separate equatorial waves into Kelvin, Rossby, MRG, and IG waves using the method proposed by Kim and Chun

(2015) with MERRA-2 reanalysis data. Here, we briefly describe the way to separate each wave component, while the details

can be found in Section 4 of Kim and Chun (2015). First, all the perturbation variables constituting the EPF are divided into

symmetric and anti-symmetric components with respect to the equator for a 90-day segment after applying sine and cosine

windows at the first and last 30 days, respectively. Second, a two-dimensional Fourier transform is performed on the

perturbation variables with respect to longitude and time to obtain their zonal wavenumber–frequency $(k - \omega)$ spectra at each

latitude and height from 30°N to 30°S and 100 hPa to 5 hPa. Kelvin waves are confined to the spectral range of $0 < k \leq 20$

and $\omega < 0.75$ cycle per day (cpd) in the symmetric spectrum in the latitude range where $|F^{z1}| < |F^{z2}|$. Here, $F^{z1}$ and $F^{z2}$

represent the first and second terms of the vertical component of EPF $(F^z)$, respectively. MRG are confined to the range $|k| \leq$

20 and $0.1 \leq \omega \leq 0.5$ cpd in the anti-symmetric spectrum within the latitude range where $F^{z1} \times F^{z2} < 0$. The spectral ranges

that are not classified as Kelvin or MRG waves, are defined as Rossby waves for the ranges $|k| \leq 20$ and $\omega \leq 0.4$ cpd, and IG

waves otherwise. Finally, EPF and EPD calculated at a given $k$ and $\omega$ are summed over the spectral range of each wave mode

using Parseval's relation (Horinouchi et al., 2003). Note that the westward waves (intrinsic frequency $< 0$) propagate in the

same direction as the EPF vectors, whereas the eastward waves (intrinsic frequency $>0$) propagate in the opposite direction of

the EPF vectors (Andrews et al., 1983). Therefore, given the dominant upward propagation in the stratosphere, EPF vectors

for the Rossby, MRG, westward IG waves are directed upward, whereas those for the Kelvin waves and eastward IG waves

are directed downward.

In the troposphere, the abovementioned method is not suitable for classifying each wave mode (Kim and Chun, 2015),

because the source of the stratospheric equatorial waves, such as convection, contaminates the $F^{z1}$ and $F^{z2}$. Therefore, we

instead apply very simple criteria to separate the wave spectrum below 100 hPa as follows: In the frequency range of $\omega \leq 0.4$

cpd, the perturbation variables in the ranges of $0 < k \leq 20$ and $-20 \leq k < 0$ are defined as low-frequency eastward $(L_e)$ and

westward $(L_w)$ waves, respectively, which approximately represent Kelvin and Rossby waves, respectively. The variables in

the spectral ranges of (i) $k > 20$ or (ii) $0 < k \leq 20$ and $\omega > 0.4$ cpd and those of (i) $k < -20$ or (ii) $-20 \leq k < 0$ and $\omega > 0.4$





cpd are defined as high-frequency eastward ($H_e$) and westward ($H_w$) waves, respectively, which approximately represent

eastward and westward IG waves, respectively. This separation method in the troposphere enables us to identify the source

location of the anomalously strong waves observed in the stratosphere.

### 2.4 Offline CGW parameterization

We apply the offline CGW parameterization using MERRA-2 data focusing on small-scale waves ($\lambda_h < 100$ km and $\lambda_z <$

40 km, where $\lambda_z$ is the vertical wavelength), which is similar to the work of Kang et al. (2017, 2018) using NCEP Climate

Forecast System Reanalysis (CFSR; Saha et al. 2010) data. The offline CGW parameterization calculates GW momentum flux

induced by convective heating rate at the source level (cloud top) as a function of phase velocity; the GW momentum flux and

drag from the cloud top to the stratosphere are calculated based on columnar wave propagation by using Lindzen's saturation

theory (Lindzen, 1981). The parameterization requires convective heating rate and convective cloud-top and -bottom heights

in addition to standard variables such as wind, temperature, and geopotential height as input data. MERRA-2 provides only

cloud-top height without convective heating rate, so we tried to extract convection-induced heating rate from the MERRA-2

output field DTDTMST; this field contains all process that contributes to latent heating by moist convection, not exclusively

by cumulus convection (Bosilovich et al., 2016). To reconcile this limitation, we select cases that satisfy several criteria to

represent clouds which can generate convective GWs. First, we only considered DTDTMST profiles in which column-

maximum height is higher than 850 hPa. Second, we estimated the convective cloud-top and -bottom heights as the locations

where DTDTMST falls to 20% and 5%, respectively, from its maximum. Here, the convective cloud top should not exceed

cloud-top height provided by MERRA-2. Although the percentage of 20% seems large, we decided to use the value considering

the large tail in the upper part of the DTDTMST profile (Fig. S1). Note that the cloud-top height is provided by MERRA-2,

but we chose instead to estimate it from the DTDTMST profile because the cloud-top heights in MERRA-2 are sometimes too

high due to stratiform clouds, such as anvil clouds, which do not represent the top height of the convection properly. Third,

when (i) the convective cloud top height is lower than 700 hPa, (ii) the convective cloud bottom height is higher than 7 km, or

(iii) the convective cloud depth is shallower than 1 km, the profiles are eliminated. The DTDTMST profiles selected using the

aforementioned procedure will be referred to as convective heating profiles hereafter; they are generally similar to the

convection-induced heating profiles estimated from satellite observations (GPM Science Team, 2017; Lang and Tao, 2018)

(Fig. S1). Note that the magnitude of the CGW momentum flux is constrained by the observed GW momentum flux from

SPBs in the tropical region (Jewtoukoff et al., 2013). Because spatio-temporal variations in convective activity and background

flows are considered in the parameterization, it is valuable to investigate the variations in the magnitude and the spectral shape

of the CGW momentum flux during the QBO disruption.





## 3. Results

**3.1 General characteristics of zonal wind and equatorial waves**

Figure 1 shows zonal-mean zonal wind in a latitude–height cross section from October 2015 to February 2016, monthly climatology from October to February, and the zonal-mean zonal wind profile averaged over 5°N–5°S during the disruption, compared with its monthly climatology. In October 2015, WQBO is very deep compared to the climatology. The WQBO starts to split into two maxima as early as November 2015, and the westerly wind becomes anomalously weak at 40–50 hPa by more

than $1\sigma$, where $\sigma$ is the standard deviation of the zonal-mean zonal wind, in December 2015. In January 2016, the zonal wind at 40 hPa continuously decelerates, and then changes into easterly in February. From January 2016, the zonal wind at the altitude above 30 hPa exhibits a strong westerly wind greater than the climatology by more than $1\sigma$, indicating that the WQBO is anomalously deep. In the upper troposphere (100–150 hPa), easterly anomalies are shown in November 2015, but from January 2016, westerly anomalies appear.

Figure 2 shows latitude–height cross sections of the EPF and EPD for each type of wave in February 2016. Note that the parameterized CGW momentum flux ($\rho_0 \overline{u'w'}$) is multiplied by ($-\cos\phi$) to display the vertical EPF vectors of CGWs, and each of the wave forcings and vectors in Fig. 2 is scaled differently in order to mainly focus on their morphology. The EPD more negative than climatology by more than $1\sigma$ is stippled, which represents anomalously strong negative wave forcing. The parameterized CGWs (P-CGWs in Fig. 2a) generally exert a positive or negative drag on the zonal wind in regions of positive

or negative wind shear, respectively, with the strongest negative forcing at 7–20 hPa between 5°N and 10°S. The negative CGW forcing at 40–50 hPa between 10°N and 10°S is anomalously strong. In 20°N–5°S, westward-propagating P-CGWs are dominant, which can be inferred from the direction of vertical EPF vectors. Kelvin waves (Fig. 2b) exert positive wave forcing in the positive shear zone, strengthening the bottom side of the westerly jet. Therefore, Kelvin waves may help to maintain two westerly jets (5–30 hPa and 50–80 hPa) with a developing easterly jet in between. MRG waves (Fig. 2c) show anomalously

strong negative forcing at 50–80 hPa, 30–40 hPa, and 15–20 hPa, concentrated at the equator. They seem to be generated in the altitude range (60–90 hPa and 30–40 hPa) in which the EPD has positive values at 5°–10°N/S. As will be shown later, the effect of the MRG waves is to flatten the meridional profile of the westerly jet, possibly making the jet more sensitive to erosion by other waves, such as Rossby waves. IG waves (Fig. 2d) near the equator (10°N–10°S) exhibit a negative forcing from 70 hPa to 5 hPa with a maximum forcing at 8–20 hPa, while the anomalously strong negative IG wave forcing is mainly

located at 50–70 hPa and 8–20 hPa. The negative Rossby wave forcing (Fig. 2e) is anomalously stronger than the climatology at 30–50 hPa between 20°N and 25°S, which is attributed to the waves that propagate from the NH extratropics. The same figure during the whole QBO disruption period from October 2015 to February 2016 is shown in Fig. S2 in the online supplemental material.

Figure 3 shows time–height cross sections of the zonal wind, zonal wind tendency, vertical advection [the second term

on the right-hand side of Eq. (1)], required wave forcing, and forcing due to each type of wave averaged over 5°N–5°S from



July 2015 to June 2016; and their monthly climatology from July to June. The required wave forcing term (REQ) is calculated as a residual by subtracting the advection terms from the zonal wind tendency in the TEM equation. In Fig. 3a, both the zonal-mean zonal wind during the disruption and the climatology propagate downward with time, but the WQBO is much deeper during the disruption than in the climatology. This feature is clearly seen in the difference plot of the zonal-mean zonal wind

(Fig. 3b), showing a strong westerly anomaly in the upper stratosphere. The westerly wind decelerates at 40 hPa from October 2015, changes into easterly in February 2016, and starts to propagate downward as an easterly QBO phase afterward. The deceleration of the westerly wind at 40 hPa is also revealed in the zonal-wind tendency (Fig. 3a), as the negative wind tendency at 40 hPa becomes anomalously strong in October 2015.

To investigate whether vertical advection contributes to the anomalous zonal wind tendency near 40 hPa, the vertical

advection term (ADVz) in the TEM equation is shown in Fig. 3d. Climatologically, the sign of the equatorial wave forcing is the same as that of the vertical wind shear. Therefore, positive $\bar{w}^*$ makes the sign of ADVz opposite to that of the vertical wind shear (Eq. 1), acting to oppose zonal wind tendency (Dunkerton, 1991). From November to December 2015 at 40 hPa, however, ADVz has the same negative sign as the zonal wind tendency because both $\bar{w}^*$ and vertical wind shear are positive (not shown), while the wave forcing is negative regardless of the positive wind shear. Therefore, ADVz acts to accelerate the easterly

development by 17% and 2% of the zonal-wind tendency, respectively, with values of -0.3 and -0.1 m s$^{-1}$ mon$^{-1}$ in November and December 2015, respectively. This implies that ADVz also contributes to the QBO disruption in the early stages.

The climatology of REQ (Fig. 3e) has negative (positive) values in the regions of negative (positive) vertical wind shear, but a sudden increase in negative REQ emerges at 40 hPa in October 2015 without negative vertical wind shear. The estimated wave forcing by P-CGWs (Fig. 3f) resembles REQ, especially at the upper stratosphere, where strong negative wind shear

exists, and in the altitude range between 40 and 70 hPa after the intrusion of easterly wind. This indicates that the P-CGWs largely contribute to the QBO disruption after the negative wind shear appears near the 40 hPa (i.e., after February 2016). Kelvin wave forcing (Fig. 3g) during the disruption is much greater than the climatology near the altitude of 30 and 60 hPa due to the positive wind shear. The wave forcing could be stronger because of the strong vertical wave flux propagating from the troposphere, which is identified by the enhanced vertical EPF for the Kelvin waves at 70 hPa (Fig. S3). The positive forcing

near 30 and 60 hPa from January to March 2016 accelerates the upper and lower jets, respectively, thereby the upper and lower parts of the QBO jet are not dissipated totally, maintaining the separated jet during the disruption (Fig. 2). Acceleration in the upper and lower parts of the separated QBO jet is also shown by the momentum forcing by P-CGWs (Fig. 3f). The contribution of CGWs to the enhanced jet in the current study may explain why the westerly winds simulated by Watanabe et al. (2018) are relatively weak compared to those in MERRA-2 near 20 hPa and 70 hPa, without non-orographic GW parameterization.

MRG wave forcing (Fig. 3h) is generally stronger during the disruption than in the climatology. In addition, there is a sudden increase in the negative MRG forcing at 40 hPa from October to November 2015, which is similar to the pattern seen in REQ at this time and location. This suggests that the MRG waves influence the early stage of the QBO disruption by slowing down the QBO jet. IG waves (Fig. 3i) exert a strong negative forcing in November 2015 contributing to the enhancement of





negative REQ near 40 hPa, together with the MRG wave forcing. Rossby wave forcing (Fig. 3j) near 40 hPa is stronger than
the climatology consistently from November 2015 to March 2016, which is considered as a major cause of the QBO disruption.

To summarize, in October 2015, the negative forcing by MRG waves is anomalously strong compared to the climatology
at 40 hPa between 5°N and 5°S, and becomes stronger in November 2015 together with IG waves when the Rossby waves
start to break at the southern hemispheric (SH) part of the QBO (see Fig. 5). Therefore, MRG and IG wave forcing may
precondition the zonal mean flow near the QBO jet core to be easily disrupted by the Rossby waves. From December 2015 to
February 2016, the Rossby wave forcing is dominant among the equatorial waves, while the negative CGW forcing contributes
significantly to the disruption in February 2016 when negative vertical wind shear appears near 40 hPa. Figure S4 is the same
figure with Fig. 3 but using ERA-I data. We found that the time evolution of each wave forcing in ERA-I is similar to that in
MERRA-2, although the magnitudes of the REQ and wave forcing (vertical advection) in ERA-I is generally stronger (weaker)
than that in MERRA-2.

**3.2 Quantitative contributions of the equatorial waves**

Figure 4 shows the time series of zonal wind, zonal wind tendency, and wave forcing by each type of wave from July
2015 to June 2016 at 40 hPa. The monthly averaged momentum forcing by each type of wave and its contribution to the total
negative wave forcing (percentage) are given in Table 1. The zonal wind (Fig. 4a) changes into easterly in February 2016,
whereas the zonal wind tendency (Fig. 4c) changes into negative value in October 2015, as shown in Fig. 3. The negative zonal
265    wind tendency in October 2015 is induced by both MRG ($-0.43$ m s$^{-1}$ mon$^{-1}$) and IG waves ($-0.46$ m s$^{-1}$ mon$^{-1}$), with
contributions of 39% and 41%, respectively, while Rossby wave forcing is $-0.22$ m s$^{-1}$ mon$^{-1}$, with a relatively small
contribution of 20% (Fig. 4b). In November 2015, negative wave forcing by Rossby, MRG, and IG waves increase with
contributions of 45%, 27%, and 28%, respectively, which are 2.4, 2.5, and 1.6 times stronger than the climatology, respectively.
Afterward, Rossby waves mainly provide negative forcing which induces easterly accelerations in December 2015 and January
270    2016, with the contributions of 70% and 91% of the total negative forcing, respectively. They are 3.2 and 4.3 times larger than
the climatology, respectively. In February 2016, Rossby waves, parameterized CGWs, MRG waves, and IG waves at 40 hPa
contribute to the total negative wave forcing by 61%, 20%, 12%, and 7%, respectively. The CGWs dominate the negative
forcing with a percentage of 60% in March 2016. When the average is taken over 5°–10°S, however, Rossby waves dominate
from October 2015 (Fig. S5). This implies that the Rossby wave forcing was strong enough to decelerate the edge of the QBO
275    jet (5°–10°S), while it presumably extends to the jet core (0°–5°S; Fig. 5) due to the weakening of the QBO jet by the MRG
(Fig. 7) and IG (Fig. 11) wave forcing near the equator.

Coy et al. (2017) showed that the positive peak of GWD in July 2015 is 4.5 m s$^{-1}$ mon$^{-1}$ and the negative GWD in February
2016 is $-0.5$ m s$^{-1}$ mon$^{-1}$ (see their Fig. 2). In the current study, the positive peak of CGW drag (CGWD) in July 2015 is 3.75
m s$^{-1}$ mon$^{-1}$ and the negative CGWD in February 2016 is $-1.0$ m s$^{-1}$ mon$^{-1}$. There are two potential reasons for this discrepancy.
First, the GWD reported by Coy et al. (2017) is provided by MERRA-2, which is based on non-orographic GWD





parameterization that does not consider GW sources explicitly. The CGWD in the current study is obtained from the physically-based CGWD parameterization, which takes into account the GW variability according to the convective activity. Second, there are some differences in analysis, such as latitude range for averaging [10°N–10°S for Coy et al., (2017) but 5°N–5°S for the present study] and the vertical grids [pressure level for Coy et al., (2017) but model level for the present study]. When we

set the average latitude as 10°N–10°S, the CGWD in February 2016 is about -0.7 m s$^{-1}$ mon$^{-1}$, which is still greater than GWD reported by Coy et al. (2017). This implies that the negative momentum forcing by CGWs is stronger than that by GWs from a fixed source during the disruption.

Fig. 4c shows the meridional and vertical components of the Rossby wave forcing at 40 hPa. The magnitude of the meridional component is larger than that of the vertical component, and becomes dominant in December, January, and

February when the negative wave forcing prevails. This demonstrates the importance of meridional propagation from extratropics, as also reported from the previous studies (e.g., Coy et al., 2016; Osprey et al. 2016). However, the vertical component is not negligible given that the maximum contribution of the vertical component to the total Rossby wave forcing reaches 26%, which is 18% of the total negative wave forcing, in December 2015 (Table 1). The strong vertical EPD in the stratosphere (40 hPa) does not necessarily indicate the wave propagation from the equatorial region. Hence, the origination of

the vertical Rossby wave forcing will be analyzed in the following subsection.

### 3.3 Contributions of Rossby waves and MRG waves

In this subsection, we focus on the Rossby and MRG waves and their sources. Figure 5 shows the evolution of the EPF and EPD for the Rossby waves (left), and their meridional (middle) and vertical (right) components, separately, from November 2015 to February 2016. The vertical profiles of meridional EPF (EPF-y) at 10°N and 10°S are included, and

meridional distribution of the vertical EPF (EPF-z) at 70 hPa are plotted at the bottom of each month in red lines. In the vertical profiles and meridional distribution plots, climatological monthly means are included with black lines with ±1$\sigma$ values indicated in gray shading. In November 2015 (Fig. 5a), the Rossby waves start to break at the southern flank of the QBO westerly jet near 50 hPa, which is anomalously strong compared to the climatology. They most likely propagate from the NH, given that EPF-y at 10°N is directed southward with a magnitude greater than the climatology by more than 1$\sigma$. The EPF-y at

10°S is directed northward at the altitude below 70 hPa, and it is slightly stronger than the climatology; however, this EPF hardly propagates into the QBO jet. In December 2015 (Fig. 5b), anomalously strong negative EPD near 40 hPa in the SH extends northward to 10°N, with the strong EPF-y at 10°N propagating toward SH. The negative EPD in the SH part of the QBO jet at 40 hPa is mainly explained by its meridional component, which presumably originates from the EPF-y at 10°N between 70 and 30 hPa. On the other hand, the negative EPD in the NH part of the QBO jet at 40 hPa is mainly explained by

its vertical component considering the anomalously strong vertical EPD there. The strong vertical EPD seems to originate from the EPF-z at 70 hPa between 0° and 15°N. In January 2016 (Fig. 5c) when the Rossby wave forcing is the strongest, the overall feature is similar to December 2015 although with somewhat different aspects: (i) negative EPD at 40 hPa exhibits a significant




peak in 0°–5°S, (ii) EPF-y at 10°N has an additional peak at 40 hPa, and the (iii) EPF-z at 70 hPa in the SH becomes much stronger than the climatology. In February 2016 (Fig. 5d), the anomalously strong negative EPD is more concentrated at 40

hPa with a larger contribution from the meridional EPD at 0°–25°N, while the vertical EPF at 70 hPa is less pronounced compared to January. To sum up, the Rossby wave forcing and the associated wave flux are anomalously strong from November 2015 to February 2016, and both the meridional and vertical components are significantly stronger than the climatology. The meridional EPD, most likely caused by waves propagating southward at 10°N, largely contributes to the deceleration of the QBO jet in the SH. The vertical EPD, presumably caused by waves propagating vertically at 70 hPa between

0° and 15°N, largely contributes to the deceleration of the QBO jet in the NH.

To investigate whether the anomalously strong EPF-z at 70 hPa in Fig. 5 originates in the equatorial region, Fig. 6 shows the EPF and EPD for the $L_w$ waves, which are westward-propagating low-frequency waves, in the troposphere and for the Rossby waves in the lower stratosphere. Here, we focus on January and February 2016 when EPF-z is strong and moderate, respectively. Note that the EPF in Fig. 6 is the same as in Fig. 2, except that it is divided by air density for better visualization.

In January 2016 (Fig. 6a), there are three potential source regions of the Rossby waves: (i) 5°N–10°S at 120–400 hPa (equatorial source), (ii) 15°–25°S at 200–350 hPa (SH source), and (iii) 20°–25°N and 250–450 hPa (NH source), considering that the positive EPD region is a source region of the westward-propagating waves. First, from the equatorial source, wave activity propagates upward and northward up to ~120 hPa. There, it seems to merge with the wave activity from the NH sources, while part of it propagates upward to the NH stratosphere near 0°–15°N. Second, some of the waves from the SH

source propagate to the SH stratosphere after depositing a large amount of negative momentum between 100 and 70 hPa, and others propagate to the NH stratosphere. Third, the wave activity from the NH source does not seem to propagate upward. In addition to the three source regions, there might be other source regions at the midlatitude, so the propagation from the midlatitudes in both hemispheres also needs to be considered. It is shown that the wave activity from the NH (SH) midlatitude propagates into the equatorial stratosphere at the altitude range above 100 hPa (200 hPa). The behaviour in February 2016

shows a similar pattern to that in January 2016, except for an additional positive EPD region at 15°–25°N and 110–150 hPa. In summary, the strong EPF-z for Rossby waves at 70 hPa is attributed to both the equatorially generated waves and the waves propagating from the NH and the SH.

Figure 7 shows the EPF and EPD for the MRG waves in October, November, and December 2015, and February 2016, when MRG waves significantly contribute to the negative EPD. The vertical profiles of EPF-y at 10°N and 10°S are plotted

on the right and left side of each panel, and the meridional distribution of the EPF-z at 70 hPa is plotted at the bottom of each panel. In Fig. 7, we will focus on the altitude near 40 hPa, where the wave forcing is directly related to the QBO disruption, although strong negative wave forcing also exists in the upper stratosphere. In October 2015 (Fig. 7a), all the wave forcing is similar to the climatology except for the MRG waves (Fig. S2); the negative MRG wave forcing is stronger than the climatology by more than $1\sigma$ at 50 hPa in 0°–5°S (indicated by the magenta dots). Given the dominant upward propagation in

the lower to middle stratosphere, the MRG waves exerting negative forcing near 50 hPa in 0°–5°S seem to propagate from 60–



80 hPa and 5°–10°S where the positive EPD exists. While the increase in the meridional EPF at 5°–10°S near 70 hPa is somewhat unclear in the EPF-y at 10°S, it is clear in the EPF-y at 7°S and 5°S showing a noticeable increase toward the equator compared to the climatology (not shown). The increase in the vertical EPF at 60–80 hPa is evident in the EPF-z at 70 hPa, which is greater than the climatology by more than $1\sigma$ in 10°S–0°. It is worthwhile to note that there exists positive EPD over

5°–10°N at 70 hPa as well, implying that 5°–10°N and 60–80 hPa might be another source region for MRG wave generation. However, the increases in the EPF-z at 70 hPa and EPF-y at 5°–10°N are less significant compared to the climatology. In November 2015 (Fig. 7b), a pattern similar to that in October 2015 appears, but with an increase in the magnitude of the negative EPD at 40–60 hPa within 5°N/S and the EPF-y at 10°S in 50–80 hPa. The strong EPD at 0°–10°S and 40–60 hPa originates most likely from the strong EPF-y at 10°S in 60–80 hPa and EPF-z at 70 hPa in 5°–15°S. In December 2015 (Fig.

7c) and February 2016 (Fig. 7d), strong negative EPD, equatorward EPF-y at 10°S, and upward EPF-z at 70 hPa are still evident.

According to this analysis, we conclude that MRG waves decelerate the QBO jet core at the onset of the QBO disruption, given that the negative zonal wind tendency from October to November 2015 is partly attributed to the anomalously strong MRG wave forcing. The positive EPD at 60–80 hPa between 5°S and 15°S by MRG is much greater than the climatology both

for the meridional and vertical components. From 60–80 hPa and 5°–15°S, the waves propagate equatorward and upward reaching 40–50 hPa near the equator (Figs. 7a–b), implying that the region of 60–90 hPa, 5°–15°S (boxed region in Fig. 8) is a possible location where the MRG waves are mainly excited.

Coy et al. (2017) have investigated whether baroclinic instability leads to the easterly wind development in February 2016, although they did not investigate the possibility of the wave generation/amplification by baroclinic instability for the

period before February 2016. As MRG waves contribute significantly to the negative EPD from October to November 2015 (Figs. 3, 4 and 7), it is worth examining whether baroclinic instability is a likely source of MRG waves (Andrews and McIntyre, 1976; Garcia and Richter, 2019) in October and November 2015, which is evaluated using the meridional gradient of the potential vorticity (Andrews et al., 1987):

$$\bar{q}_\phi = 2\Omega \cos\phi - \left[\frac{(\bar{u}\cos\phi)_\phi}{a\cos\phi}\right]_\phi - \frac{a}{\rho_0}\left(\frac{\rho_0 f^2}{N^2}\bar{u}_z\right)_z, \tag{3}$$

where $\Omega$ is the Earth's rotation and $N$ is the buoyancy frequency. The negative regions of $\bar{q}_\phi$ suggest a possibility of baroclinic instability, because the positive and negative $\bar{q}_\phi$ values in a neighboring region satisfy the necessary condition for the instability (Gill, 1982).

Figure 8 presents the monthly-mean $\bar{q}_\phi$ (Figs. 8a–b) and the number of grids where the daily-mean $\bar{q}_\phi$ is negative (Figs. 8c–d) in October and November 2015. The monthly-mean $\bar{q}_\phi$ in the boxed region has small positive values in October 2015 (Fig. 8a) and November 2015 (Fig. 8b). In Figs. 8c–d, however, 22 and 25 days during October and November, respectively,





have negative $\bar{q}_\phi$ values by at least one point within the boxed region, which satisfy a necessary condition for baroclinic instability, dominated by the barotropic instability (not shown). The number of negative $\bar{q}_\phi$ days are much larger than the

climatology (11 and 19 days with standard deviations of 11 and 7 days in October and November, respectively; not shown). This suggests that baroclinic instability at the boxed region is a possible source for generating the anomalously strong MRG waves. The MRG waves generated by the baroclinic instability in the narrow westerly jets accelerate the zonal wind off the equator and decelerate the zonal wind near the equator, reducing the curvature and thus the instability, which indicates that the MRG waves respond to the QBO wind system (Garcia and Richter, 2019). However, as the deceleration of the jet core is

important in the QBO disruption, such behavior may play an important role in preconditioning the background wind. It should be noted that the baroclinic instability does not seem to be an exclusive source of the MRG waves because there exist precedent WQBO cases having considerable negative $\bar{q}_\phi$ without significant enhancement in the wave generation (e.g., 2010, 1987, and 1982). Therefore, further studies on the source of the MRG waves should be done in the future. It is also interesting that $\bar{q}_\phi$ shows large negative values at the upper stratosphere (~5 hPa) where the zonal wind curvature is large in association with

strong westerly jet (Hamilton, 1984). Nevertheless, it is unlikely that the MRG waves generated at 5 hPa affects the QBO disruption as the upward propagating MRG waves (i.e., vertical EPF >0) are dominant in the stratosphere, and the strong easterlies between 5 to 10 hPa inhibit the propagation of the MRG waves.

    We found in the previous figures (Figs. 5 and 6) that the increased Rossby wave forcing in the stratosphere partly originates from the equatorial troposphere. Therefore, in Fig. 9, we examine the zonal-mean precipitation in the equatorial

troposphere to identify convective activity using MERRA-2 data. Overall, the zonal-mean precipitation from November 2015 to February 2016 is stronger than the climatology in 5°N–5°S. It is greater than the climatology by more than $1\sigma$ from November to December 2015 (Figs. 9a–b). In February 2016 (Fig. 9d), the precipitation is much stronger than the climatology between 5°N and 10°S by more than $3\sigma$. The maximum precipitation is slightly shifted southward in December–January–February (DJF), following the location of the inter-tropical convergence zone (ITCZ).

We further check whether the precipitation spectrum related to each equatorial wave type is enhanced during the disruption. Figure 10 illustrates the power spectrum of the precipitation data of MERRA-2 divided by background spectrum averaged over 10°N–10°S for both the symmetric and anti-symmetric components. The background spectrum is obtained by applying 1-2-1 smoothing to the base-10 logarithm of the raw spectrum (separately for the symmetric and antisymmetric spectrum) in wavenumber and frequency 40 and 10 times, respectively, and applying based-10 exponential again to the

smoothed spectrum (Chao et al., 2009). If the raw spectrum divided by the background spectrum is greater than 1.4, it is considered statistically significant at the 95% level for 41 degrees of freedom (i.e., corresponding to the number of the latitude grid cells from 10°N to 10°S) (Wheeler and Kiladis, 1999).

    The area where the precipitation spectrum is greater than the climatology by more than $1\sigma$, which is denoted by a stippled pattern, widens from November 2015 to February 2016, indicating that not only the mean value but also the variability of the

convection significantly increases during the disruption. The spectra related to Rossby waves in the symmetric spectrum ($k =$





-10–0, $\omega$ = 0–0.15 cpd) are statistically significant throughout the period, suggesting that the convective activity in the troposphere is the probable source for Rossby waves. However, the waves in the low-frequency spectra have less possibility to propagate upward into the stratosphere due to their slow vertical-group velocity (Yang et al., 2011). In November 2015 (Fig. 10a) and December 2015 (Fig. 10b), the spectra related to MRG waves in the anti-symmetric component ($k$ = -9 and $\omega$ = 0.12;

$k$ = -5 and $\omega$ = 0.28) are statistically significant and their amplitude is stronger than the climatology by more than $1\sigma$. However, they are less likely to be the primary source of the anomalously negative MRG wave forcing in the stratosphere, given that the EPF-z for MRG waves greater than the climatology only appears at altitude above 70 hPa (Fig. 7). It is also interesting that the peaks related to Kelvin waves ($k$ = 0–10 and $\omega$ = 0–0.25) are increasing from November 2015 (Fig. 10a) to February 2016 (Fig. 10d), consistent with the increasing EPF-z at 70 hPa (see Fig. S3) and the resultant EPD (Fig. 2) during the disruption.

To validate the realism of the MERRA-2 precipitation data, we additionally calculate the space-time spectra of precipitation provided by TRMM in Fig. S6. The key features are present in TRMM, but its amplitude is larger than that of MERRA-2, possibly attributed to the finer resolution. Note that TRMM data are available in a shorter period (1998–2016) than the MERRA-2 data (1980–2016), so only five years are included as WQBO climatology in Fig. S6. As in the MERRA-2 data, there exists significant increase in the precipitation of TRMM during January and February 2016 (Fig. S6) compared to

the climatology.

### 3.4 Contribution of inertia-gravity waves

Figure 11 shows EPF vectors and EPD for the IG waves, together with the meridional distribution of EPF-z at 70 hPa, from November 2015 to February 2016. In this figure, the vertical cross section of EPF-y is not shown because EPF-z dominates the total EPF. EPF-z here is the net EPF of eastward and westward IG waves, so the positive EPF-z indicates

stronger westward EPF than the eastward one, given the dominant upward propagation in the stratosphere. Anomalously strong negative wave forcing exists at 10–20 hPa near the equator throughout the period. In November 2015 (Fig. 11a), negative wave forcing is anomalously strong at 40–80 hPa near the equator (0°–5°S), influencing the deceleration and the downward shift of the WQBO jet core in the following months. The strong negative forcing is likely attributable to the strong vertical EPF at 70 hPa, which is greater than the climatology by more than $1\sigma$. In December 2015 (Fig. 11b) and January 2016 (Fig. 11c), it is

shown that the negative wave forcing at 40–80 hPa near the equator and the westward EPF-z at 70 hPa in 10°N–10°S are anomalously strong, as in November 2015. In February 2016, negative wave forcing exists at 40 hPa and 0°–5°S without significance, while the negative wave forcing near the top of the lower jet is significant. From November 2015 to February 2016, the strong westward IG wave forcing is mainly induced by the vertical EPF penetrating into the stratosphere from the troposphere. Then why do the westward IG waves at 70 hPa show a noticeable increase during the disruption?

To answer this question, we show the vertical EPF for $H_e + H_w$ waves [(i) $|k| > 20$ for all $\omega$ or (ii) $|k| \leq 20$ for $\omega > 0.4$ cpd] (approximately for IG waves) at the source level from November 2015 to February 2016 in Fig. 12, where the source level is set to 140 hPa based on the changes in the sign of EPD (not shown). The eastward and westward waves have similar





magnitudes in November 2015. However, the westward-propagating waves start to dominate the vertical EPF from December 2015. In January and February 2016, the vertical EPF at 140 hPa is greater than the climatology by more than $1\sigma$ at 10°N–

10°S. The stronger westward EPF-z at 140 hPa from December 2015 to February 2016 suggests that the preference for westward propagating waves at 70 hPa stems from the source level except in November 2015.

Figure 13 illustrates the power spectral density of the precipitation in a phase-speed spectrum of the $H_e + H_w$ waves from November 2015 to February 2016 along with the climatology. The precipitation spectrum is classified as eastward- (westward-) propagating waves when the phase-speed is larger (smaller) than the zonal wind at the source level. The double-sided arrows

represent the zonal wind range between the source level (140 hPa) and 70 hPa in each month, indicating the phase-speed range of the critical-level filtering. In November 2015 (Fig. 13a), the zonal wind at the source level is near zero, so the precipitation spectrum has a similar amplitude between the eastward and westward waves. However, the eastward waves are almost filtered out due to the positive vertical wind shear between 140 and 70 hPa (see Fig. 1). This feature is different from the climatology, which has stronger westward waves than eastward waves at the source level. As most of the pronounced westward waves are

filtered out due to the negative vertical wind shear (see Fig. 1), the remaining spectrum at 70 hPa in November 2015 has more westward waves than the climatology. In December 2015 (Fig. 13b), the wave characteristics at the source level during the disruption agree well with the climatology—that is, stronger westward waves than eastward waves and a similar magnitude of westerly winds at the source level. However, both a larger magnitude of the precipitation spectrum and the narrower critical-level filtering range for the westward waves result in stronger westward momentum flux at 70 hPa during the disruption than

the climatology. From January 2016 (Fig. 13c), (i) the westerly anomaly at the source level, (ii) strong precipitation spectrum, and (iii) decreased critical-level filtering of the westward waves induce a stronger westward momentum flux at 70 hPa. The presence of stronger westerlies at the source level than the climatology during the disruption becomes apparent in February 2016, leading to the strongest westward momentum flux at the source level. Figure 13 suggests that strong westward IG waves at 70 hPa during the disruption are largely attributed to the reduced critical-level filtering of westward waves in November

2015, while in December 2015, those are attributed to both enhanced convection and the decreased critical-level filtering. In January and February 2016, westerly anomaly at the source level and the reduced filtering of westward waves plays an important role in the increased westward IG wave forcing, along with stronger convection.

Kawatani et al. (2019) showed stronger westward wave forcing between 40 hPa and 70 hPa during El Niño than during La Niña in their Model for Interdisciplinary Research on Climate-Atmospheric General Circulation Model (MIROC-AGCM)

simulation. They explained that larger westward forcing is due to the strong westward EPF in the UTLS, which is attributed to the enhanced convective activity with $-10 < c < 10$ m s$^{-1}$ (where $c$ is the phase speed) and less critical-level filtering of the IG waves during El Niño than during La Niña. This is consistent with our result, implying that the enhanced wave source (i.e., convection) and the propagation conditions favorable for westward IG waves in the current study are presumably associated with the strong El Niño condition.



## 3.5 Contribution of parameterized CGWs

Figure 14 illustrates the zonal-mean zonal CGWD overlaid with zonal-mean zonal wind profile and the source-level CGW momentum flux averaged over 5°N–5°S in February 2016, when CGWD started to contribute to the QBO disruption, along with the climatology. Negative CGWD appears where the vertical wind shear is negative, with a maximum magnitude of -1.9 m s$^{-1}$ mon$^{-1}$ at 47 hPa. Once a negative vertical wind shear develops, CGWs begin to exert negative forcing on the zonal wind, making the vertical wind shear stronger, which in turn leads to a greater negative CGWD. It is noticeable that the source-level CGW spectrum reveals much stronger momentum flux than the climatology and the difference from the climatology is larger for the westward momentum flux than the eastward momentum flux, resulting in a faster and more irreversible easterly development at 40 hPa. In addition to the source spectrum, the apparent positive wind shear in the upper troposphere during February 2016 enhances the propagation of westward waves into the stratosphere in comparison to the negative wind shear in the climatology.

We would like to answer the following two questions: (1) Why is the source-level CGW momentum flux stronger in February 2016 than in the climatology? (2) Why is the increased amount of westward momentum flux larger than that of eastward momentum flux? Figure 15 illustrates the convective source spectrum and the wave-filtering and resonance factor (WFRF) spectrum, which are two important factors constituting source-level CGW momentum flux spectrum in the parameterization by Kang et al. (2017). The magnitude of convective source spectrum is proportional to the square of the convective heating rate, having a peak where the phase speed equals to the moving speed of convection ($c_{qh}$). WFRF includes two main effects: (i) critical-level filtering within the convective forcing region and (ii) the amplification of the response due to the matching of the vertical wavelength of the GW and the vertical configuration of convective heating. As the convective heating is deeper, WFRF integrated over all phase speeds becomes larger and its peak is shifted to the higher phase speed (Song and Chun, 2005). The magnitude of the convective source spectrum (Fig. 15a) is much stronger than the climatology and WFRF (Fig. 15b) shows a stronger magnitude throughout all phase speeds, both of which lead to the exceptionally strong momentum flux of CGWs. Stronger magnitude of WFRF is due to a higher static stability and deeper convection that are possibly triggered by El Niño, in which there is a warm troposphere and cool stratosphere (Domeisen et al., 2019; Kawatani et al., 2019; Richter et al., 2020). The zonal wind at the cloud top (white line) exhibits a weaker easterly compared to the climatology (gray line): zonal winds at the cloud top averaged over 5°N–5°S are -3.4 and -4.4 m s$^{-1}$ for the disruption and the climatology, respectively. On the other hand, the difference in $c_{qh}$ (gray line) is negligible. Thus, the westerly wind anomaly at the cloud top is responsible for the westward CGWs that are increased more than the eastward CGWs at the source level during the disruption.

## 4. Summary and Conclusion

In this study, we have investigated the contribution of each equatorial planetary wave mode and parameterized convectively-excited gravity waves, CGWs, to the 2015–16 QBO disruption by utilizing the equatorial wave separation method



of Kim and Chun (2015) and the offline CGW parameterization by Kang et al. (2017) using MERRA-2 model-level data. The main results, represented in schematic form in Fig. 16, are as follows:

- From October to November 2015, anomalously strong negative forcing by MRG waves mainly decelerated the QBO jet at 0°-5°S near 40–50 hPa. From November 2015, IG wave forcing became anomalously strong at the altitudes below 50 hPa, when the Rossby waves propagating from the NH began to break at the southern flank of the westerly jet (0°-10°S) at 30–60 hPa. The anomalous MRG waves were possibly generated by the increased frequency of barotropic instability in the lower stratosphere. IG wave forcing was attributed to (i) stronger convection in the equatorial troposphere, (ii) stronger westerly (or weaker easterly) winds leading to an enhanced westward momentum flux at the source level, and (iii) the reduced critical-level filtering of the westward waves arising from weaker negative wind shear in the UTLS compared to the climatology.

- From December 2015, Rossby-wave breaking extends from the SH to the equator. The deceleration of the QBO jet in the NH was mainly induced by the vertically propagating Rossby waves penetrating into the stratosphere. They likely originated in the NH and SH extratropics as well as in the tropics, generated by the convection in the equatorial troposphere. The deceleration of the QBO jet in the SH is mainly induced by Rossby waves propagating laterally from the NH extratropics. In January 2016, Rossby wave forcing was the strongest among all equatorial waves.

- In February 2016, the QBO jet at 40 hPa was continuously decelerated by the Rossby waves, propagating both vertically and latitudinally. At the same time, the estimation of the CGW forcing suggests that CGWs provided negative forcing on the QBO jet at 40–50 hPa near the equator, contributing to 20% of the total negative wave forcing. The enhancement in the negative CGWD is partly explained by excessively strong westward momentum flux at the source level, which was attributed to the westerly wind anomaly at the source level and the reduced critical-level filtering of the westward waves in the upper troposphere.

- Meanwhile, the Kelvin waves and CGWs helped confine the development of the easterlies to the region near 40 hPa by strengthening the westerly jets near 20–30 hPa and 60–80 hPa from January 2016.

In previous studies, laterally propagating Rossby waves from the midlatitudes have been considered as the primary cause of the QBO disruption, although Lin et al. (2019) emphasized the role of local equatorial wave forcing in preconditioning the Rossby wave breaking. In the present study, we found that anomalously strong negative MRG and IG wave forcing in the early stage of the QBO disruption played a significant role in preconditioning the QBO jet core. Figure 17 shows scatter plots demonstrating how the wave flux or wave forcing was anomalously strong compared to the climatology. The negative EPDs for the MRG and IG waves in 2015 were the strongest among those in other WQBO cases (Fig. 17a), where the EPDs for the MRG waves and IG waves are averaged for October–November and November–December, respectively. We also found that Rossby waves propagating upward from the equatorial troposphere significantly contribute to the QBO jet in the NH, which helped to interrupt the westerly jet along with the equatorward propagating Rossby waves. Both the meridional and vertical EPF of the Rossby waves propagating into the equatorial stratosphere averaged for January–February in 2016 were stronger





than those in any other years of WQBO phases (Fig. 17b). The contribution of the parameterized CGWs to the QBO disruption, which had been considered small, was found to be substantial when a physically based CGW parameterization was used; the negative CGWD in February 2016 was the largest among CGWD values in February with WQBO phases at 30–50 hPa (Fig. 17c). The strongest CGWD at 30–50 hPa is not surprising given that 2016 is the only year when the negative vertical wind shear occurs near 40 hPa due to sudden easterly development. However, it is surprising that the westward CGW momentum

flux at the source level in February 2016 was much stronger than in any of the February with WQBO phases (Fig. 16c). This suggests that the variability of the GWs according to the convective activity leads to an enhancement in the negative CGWD at 40 hPa.

The current results are based on the MERRA-2 data, and some uncertainties might be included in association with reanalysis data. Therefore, we checked whether the behavior of the equatorial waves in MERRA-2 also appears in ERA-I

during the QBO disruption period (Fig. S3). The equatorial wave forcing in ERA-I showed similar time evolution to that in MERRA-2, despite somewhat larger wave forcing in ERA-I. In addition, the tropical precipitation in MERRA-2, which increased during the QBO disruption, was found to be evident in the observed precipitation (TRMM; Fig. S6). One additional point to mention about the uncertainties in our results is on the cloud top and bottom heights used for the CGW parameterization. Although we tried to make vertical profiles of the convective heating rate comparable to those estimated from the satellite

observations (Sect. 2.4), CGW momentum flux spectrum is very sensitive to the cloud-top and -bottom heights (Song and Chun, 2005; Kang et al., 2017) that are derived from the threshold percentage of the convective heating profiles. Considering the importance of cloud-top and -bottom heights, their realistic estimation needs to be further investigated in the future.

Although not discussed in the results section, a QBO westerly phase that does not rapidly propagate downward and maintains westerly winds throughout a deep layer might provide a favorable condition for the QBO disruption. Hitchcock et

al. (2018) mentioned that westerly QBO should be deep enough to develop easterly winds away from the top and bottom shear regions of the jet. In addition, Osprey et al. (2016) reported enhanced tropical upwelling during the disruption. In our analysis, it is found that not only the mean upwelling ($\overline{w}^*$) in the whole stratosphere was strengthened but also upwelling in the upper stratosphere was stronger than the climatology (c.f. strong positive ADVz at the top of the QBO in Fig. 3), which made a deep and stalled QBO jet susceptible to the continuous deceleration by wave forcing. Therefore, it would be interesting to investigate

the vertical upwelling and its importance during the disruption period.

In this study, we found that the 2015–16 QBO disruption occurred when the following conditions were met: (i) negative equatorial wave (MRG, IG) forcing in the early stages and (ii) strong vertical and horizontal components of Rossby waves with strong small-scale CGWs in the later stages. The enhancement in the convective activity as well as anomalous wind profile, possibly attributed to a strong El Niño, leads to anomalously strong negative equatorial wave forcing. However, it is

still puzzling why the equatorial wave activity in 2015/16 is stronger than those in other El Niño periods, which requires further investigation. Because more frequent occurrences of the QBO disruption are expected in a warmer climate (Osprey et al., 2016;



Hirota et al., 2018), understanding the 2015–16 QBO disruption would eventually lead to an improvement of the long-range forecast in the future.

*Data availability.* The MERRA-2 data were provided by the Global Modeling and Assimilation Office at NASA Goddard Space Flight Center through the NASA GES DISC online archive (available online at https://gmao.gsfc.nasa.gov/reanalysis/). The ERA-I data were obtained from the ECMWF data server (available online at http://apps.ecmwf.int/datasets/).

*Author contributions.* HYC and MJK designed the study and MJK carried it out. MJK prepared the manuscript with
contribution from HYC and RRG. All co-authors interpreted the results and reviewed and edited the paper.

*Competing interests.* The authors declare they have no conflict of interest.

*Acknowledgments.* This work was supported by the National Research Foundation of Korea(NRF) grant funded by the Korea
government(MSIT) (No. 2020R1A4A1016537). We would like to thank J.-H. Yoo for helpful comments.

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



**Table 1.** Momentum forcing at 40 hPa by each wave (m s$^{-1}$ mon$^{-1}$) and its percentage to the total negative wave forcing (parenthesis) averaged over 5°N–5°S from October 2015 to March 2016 and for the climatology. The percentage is calculated when a wave forcing is negative during the QBO disruption.

| 2015-16 | Oct 2015 | Nov 2015 | Dec 2015 | Jan 2016 | Feb 2016 | Mar 2016 |
|---|---|---|---|---|---|---|
| **MRG** | -0.4 (39%) | -0.6 (27%) | -0.6 (17%) | -0.1 (1%) | -0.6 (12%) | -0.5 (10%) |
| **IG** | -0.5 (41%) | -0.6 (28%) | -0.5 (13%) | -0.4 (8%) | -0.3 (7%) | -0.4 (7%) |
| **Rossby** | -0.2 (20%) | -0.9 (45%) | -2.6 (70%) | -4.4 (91%) | -3.0 (61%) | -1.3 (23%) |
| **CGW** | 0.8 | 0.6 | 0.3 | 0.5 | -1.0 (20%) | -3.3 (60%) |
| **Kelvin** | 0.9 | 0.6 | 0.7 | 1.1 | 0.7 | 0.5 |
| **Rossby-Y** | -0.3 (30%) | -0.7 (34%) | -1.9 (52%) | -3.8 (78%) | -2.7 (55%) | -1.4 (25%) |
| **Rossby-Z** | 0.03 | -0.2 (11%) | -0.7 (18%) | -0.6 (13%) | -0.3 (6%) | 0.1 |
| **Climatology** | Oct | Nov | Dec | Jan | Feb | Mar |
| **MRG** | -0.3 | -0.2 | -0.2 | -0.2 | -0.1 | -0.1 |
| **IG** | -0.2 | -0.4 | -0.4 | -0.4 | -0.3 | -0.4 |
| **Rossby** | -0.3 | -0.4 | -0.8 | -1.0 | -1.1 | -0.8 |
| **CGW** | 1.0 | 0.6 | 0.5 | 0.5 | 0.5 | 0.5 |
| **Kelvin** | 1.6 | 1.2 | 1.0 | 1.0 | 1.0 | 0.9 |
| **Rossby-Y** | -0.2 | -0.3 | -0.6 | -0.8 | -0.9 | -0.6 |
| **Rossby-Z** | -0.1 | -0.1 | -0.2 | -0.2 | -0.2 | -0.2 |





**Figure 1.** (Top) Latitude–height cross sections of the zonal-mean zonal wind from October 2015 to February 2016, (middle) their climatology in each month from October to February, and (bottom) those averaged between 5°N and 5°S. Red lines
represent the QBO disruption and black lines represent the climatology with ± 1 standard deviation (gray shading).









**Figure 2.** Latitude–height cross sections of the EPF (vectors) and EPD (shading) for the (a) parameterized CGWs (P-CGWs, multiplied by 2) and resolved equatorial waves, including (b) Kelvin (multiplied by 2), (c) MRG (multiplied by 4), (d) inertia-gravity (IG, multiplied by 4), and (e) Rossby waves, superimposed on the zonal-mean zonal wind (contour) in February 2016. Positive (negative) zonal winds are plotted with solid (dashed) lines with a contour interval of 2 m s$^{-1}$, and thick contour lines denote a zero zonal wind speed. The magenta stippled pattern represents a region where the EPD is algebraically smaller (more negative) than the climatology by more than its standard deviation. The arrow on the upper right corner denotes the reference vector.



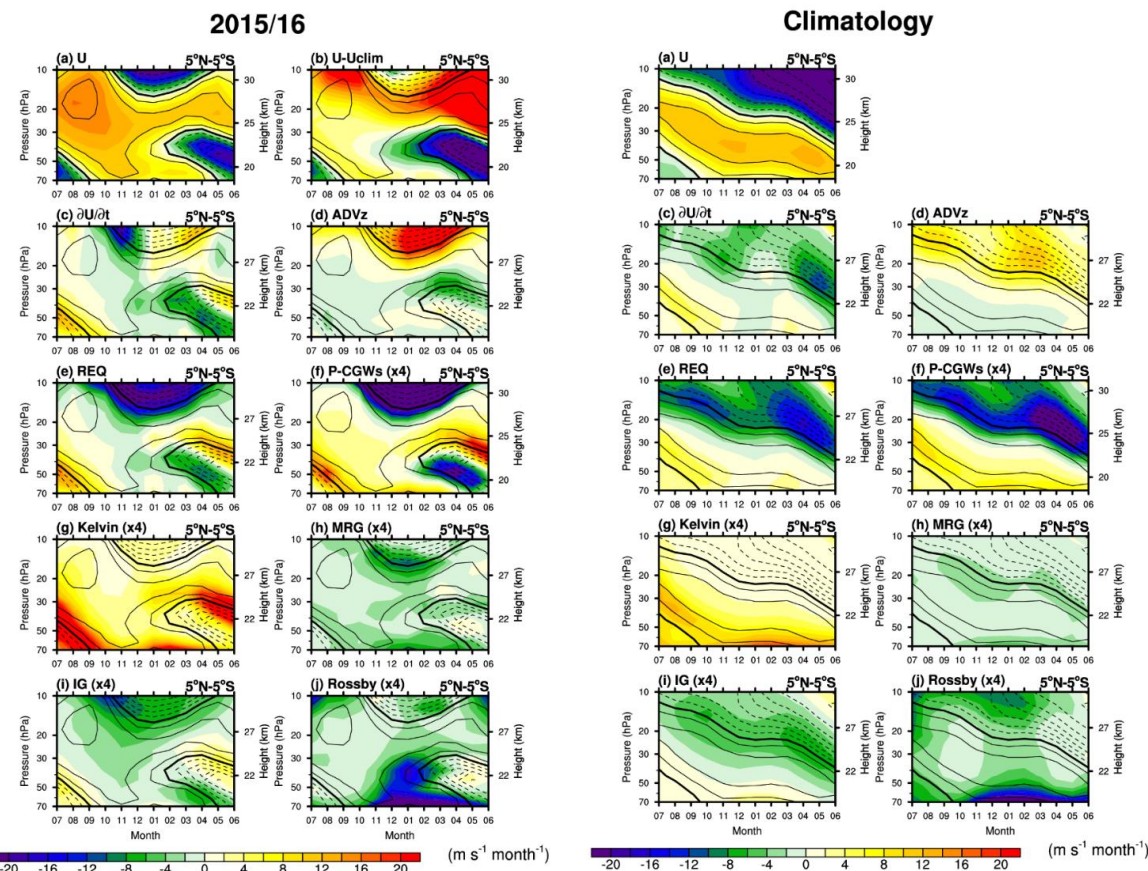

**Figure 3.** Time–height cross sections of the (a) zonal-mean zonal wind (U), (b) zonal wind anomaly from the climatology (U-Uclim), (c) zonal wind tendency ($\partial U/\partial t$), (d) vertical advection (ADVz), (e) required wave forcing (REQ) in the TEM equation, and EPD for the (f) P-CGWs, (g) Kelvin, (h), MRG, (i) IG, and (j) Rossby waves (left) from July 2015 to June 2016 and (right) their climatology from July to June, overlaid with the zonal-mean zonal wind (black contour lines). Positive (negative) zonal winds are plotted with solid (dashed) lines with a contour interval of 5 m s$^{-1}$, and thick contour lines denote a zero zonal wind speed.

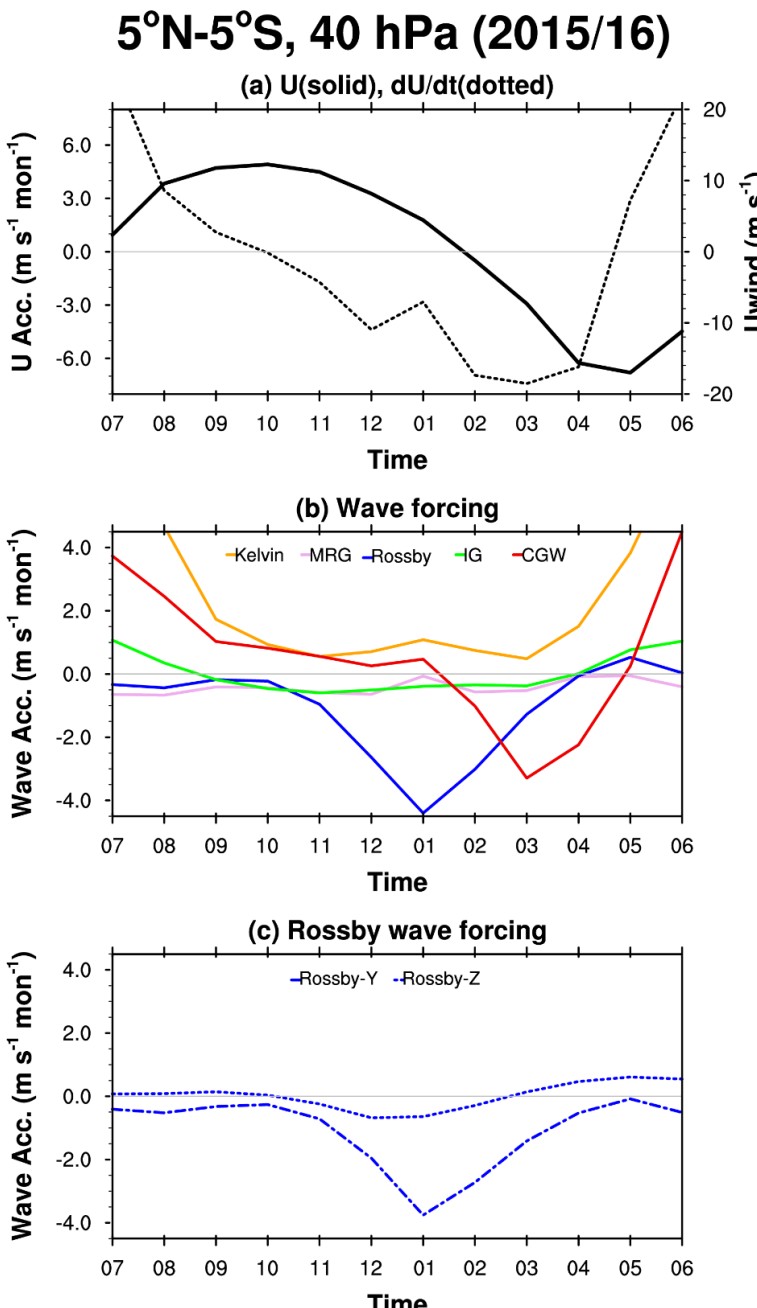

**Figure 4.** Time series of the (a) zonal-mean zonal wind (solid) and zonal wind tendency (dotted), (b) wave forcing by the
Kelvin waves (orange), MRG waves (pink), Rossby waves (blue), IG waves (light green), and CGWs (red), and (c) meridional
(dot-dashed) and vertical components (dotted) of the Rossby wave forcing, averaged over 5°N–5°S at 40 hPa from July 2015
to June 2016.







**Figure 5.** Latitude–height cross sections of the (1st column) EP flux (vectors) and EP flux divergence (EPD, shading) for the Rossby waves, (2nd column) their meridional component, and (3rd column) their vertical component in (a) November 2015, (b) December 2015, (c) January 2016, and (d) February 2016. The panel on the left (right) side of the meridional component represents the meridional EP fluxes at 10°S (10°N) and the panel under the vertical component represents the vertical EP flux at 70 hPa [red lines for each month and black lines for their monthly climatology with ±1 standard deviation (gray shading)]. Positive (negative) zonal winds are plotted with solid (dashed) lines with a contour interval of 2 m s$^{-1}$, and thick contour lines denote a zero zonal wind speed. The magenta stippled pattern represents a region where the EPD is smaller than the climatology by more than its standard deviation.

785



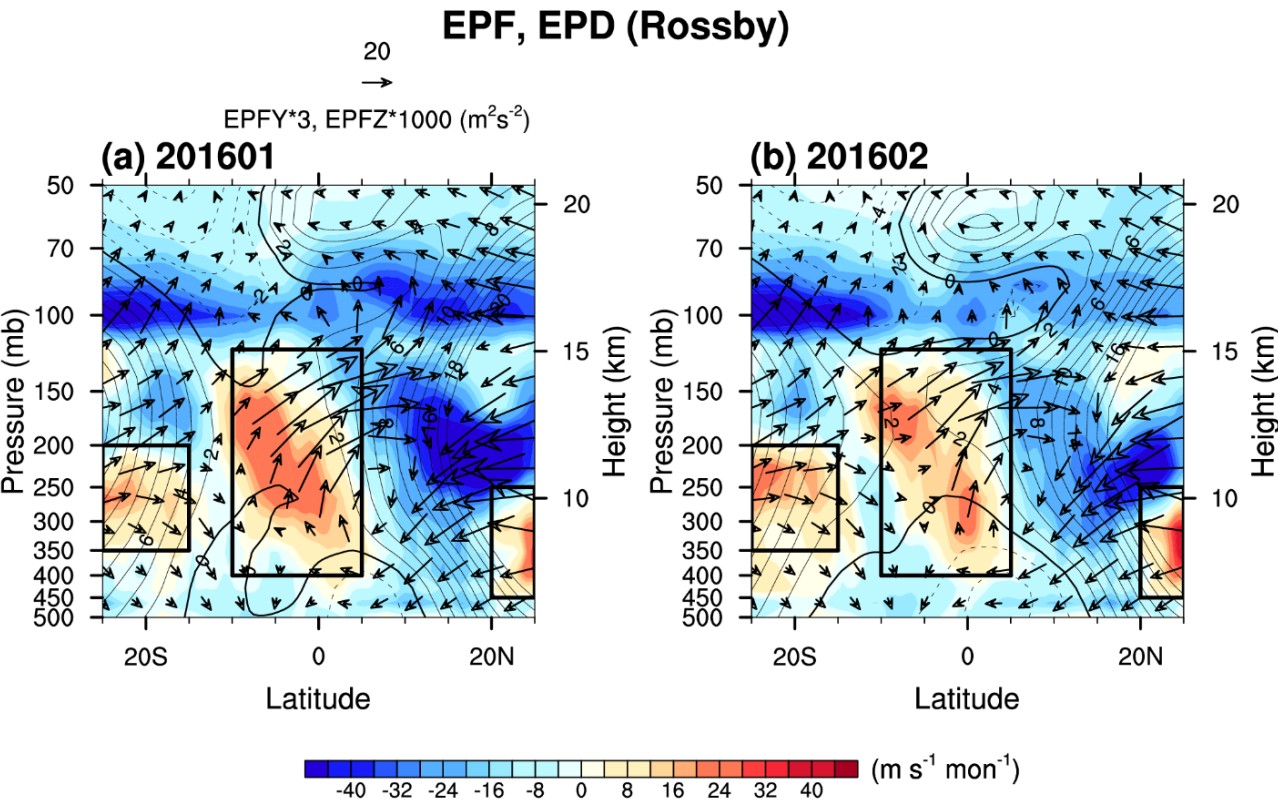

**Figure 6.** Latitude–height cross sections of the EP flux (vectors) and EP flux divergence (shading) for the Rossby waves and in (a) January 2016 and (b) February 2016. Note that below 100 hPa, the $L_w$ waves ($\omega \leq 0.4$ cpd and $-20 \leq k < 0$) are assumed to be the Rossby waves. Black boxes denote the three-potential source regions.



**Figure 7.** Latitude–height cross sections of the EP flux (vectors) and EP flux divergence (EPD, shading) for the MRG waves, in (a) October 2015, (b) November 2015, (c) January 2016, and (d) February 2016. The panel on the left (right) side of the EPD represents the meridional EP fluxes at 10°S (10°N) and the panel under the EPD represents the vertical EP flux at 70 hPa [red lines for each month and black lines for their monthly climatology with ±1 standard deviation (gray shading)]. Positive (negative) zonal winds are plotted with solid (dashed) lines with a contour interval of 2 m s⁻¹, and thick contour lines denote a zero zonal wind speed. The magenta stippled pattern represents a region where the EPD is smaller than the climatology by more than its standard deviation. Here, EPF and EPD are multiplied by 8 and 4, respectively.





**Figure 8.** Latitude–height cross sections of the monthly-mean baroclinic instability (shading) superimposed on the zonal-mean zonal wind (contour) in (a) October 2015 and (b) November 2015. Positive (negative) zonal winds are plotted with solid (dashed) lines with a contour interval of 2 m s⁻¹, and thick contour lines denote a zero-zonal wind speed. Daily time series of the number of grids where daily-mean $\overline{q}_\phi$ is negative in the boxed region (5°–15°S, 60–90 hPa) in (c) October 2015 and (d) November 2015.


**Figure 9.** Zonal-mean precipitation of MERRA-2 in (a) November 2015, (b) December 2015, (c) January 2016, and (b)
February 2016 (red) and their monthly climatology (black) with ± 1 standard deviation (gray shading).

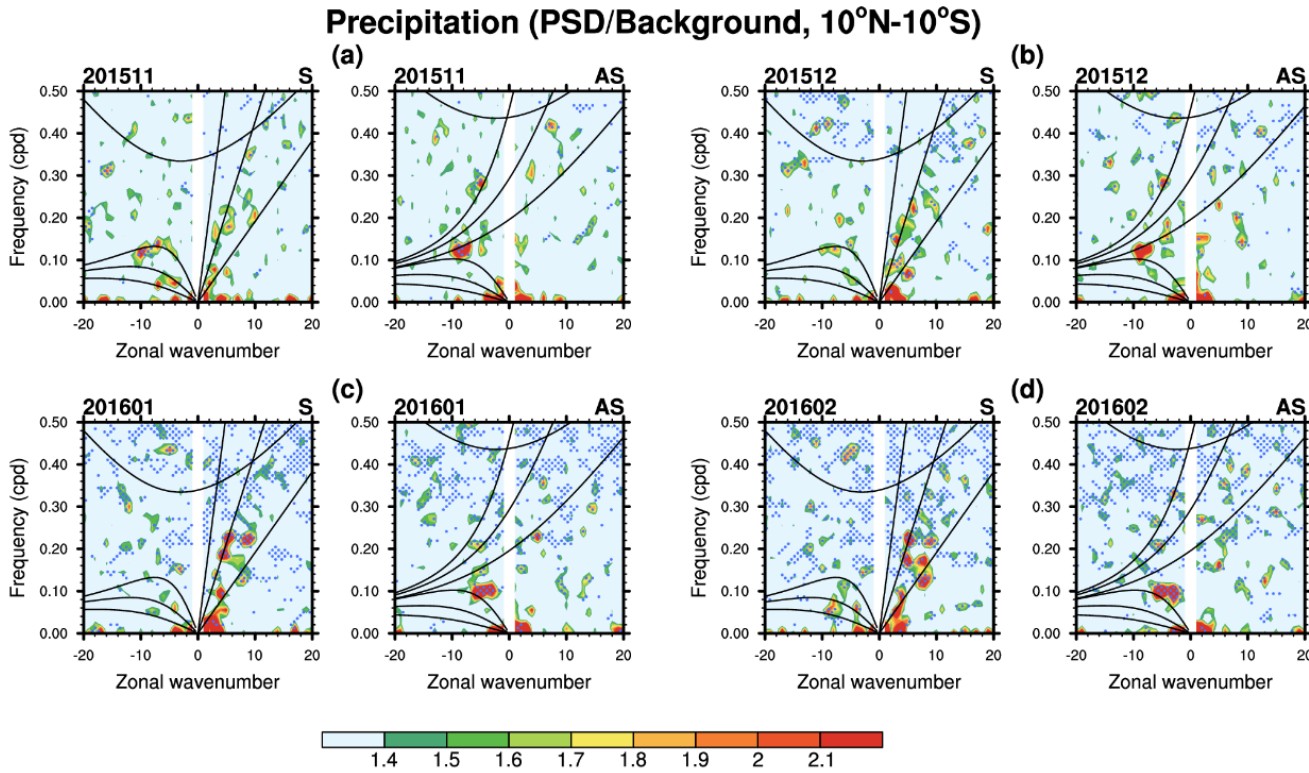

**Figure 10.** Power spectral density of the MERRA-2 precipitation divided by the background spectrum (see text for details) as function of zonal wavenumber and frequency for (left) symmetric and (right) antisymmetric components averaged over 10°N–10°S in (a) November 2015, (b) December 2015, (c) January 2016, and (d) February 2016. When the ratio between the raw power and the background power is larger than 1.4, it is considered a statistically significant spectrum at 95% level. The blue-stippled pattern denotes a spectrum where the power is stronger than the climatology by more than its standard deviation. Thick solid lines denote theoretical dispersion lines of each equatorial waves for the equivalent depth of $h$ = 8, 40, 240 m, although only $h$ = 8 m line is shown for IG waves.



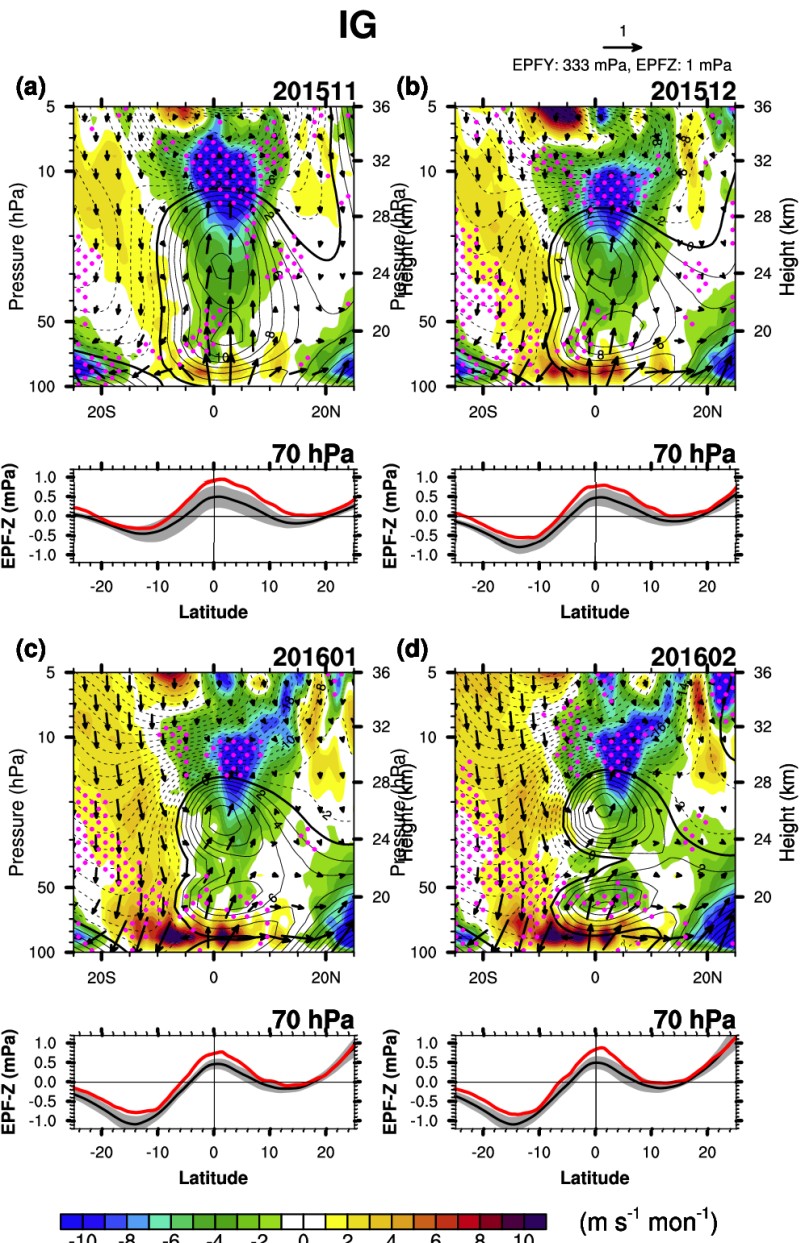

**Figure 11.** Latitude–height cross sections of the EP flux (vectors) and EP flux divergence (shading) for the IG waves (multiplied by 4) with the (bottom) vertical EP flux for the IG waves at 70 hPa in (a) November 2015, (b) December 2015, (c) January 2016, and (d) February 2016 (red) and their monthly climatology (black) with ±1 standard deviation (gray shading). Positive (negative) zonal winds are plotted with solid (dashed) lines with a contour interval of 2 m s$^{-1}$, and thick contour lines denote a zero zonal wind speed. The magenta stippled pattern represents a region where the EPD is smaller than the climatology by more than its standard deviation. Here, EPF and EPD are multiplied by 8 and 4, respectively.


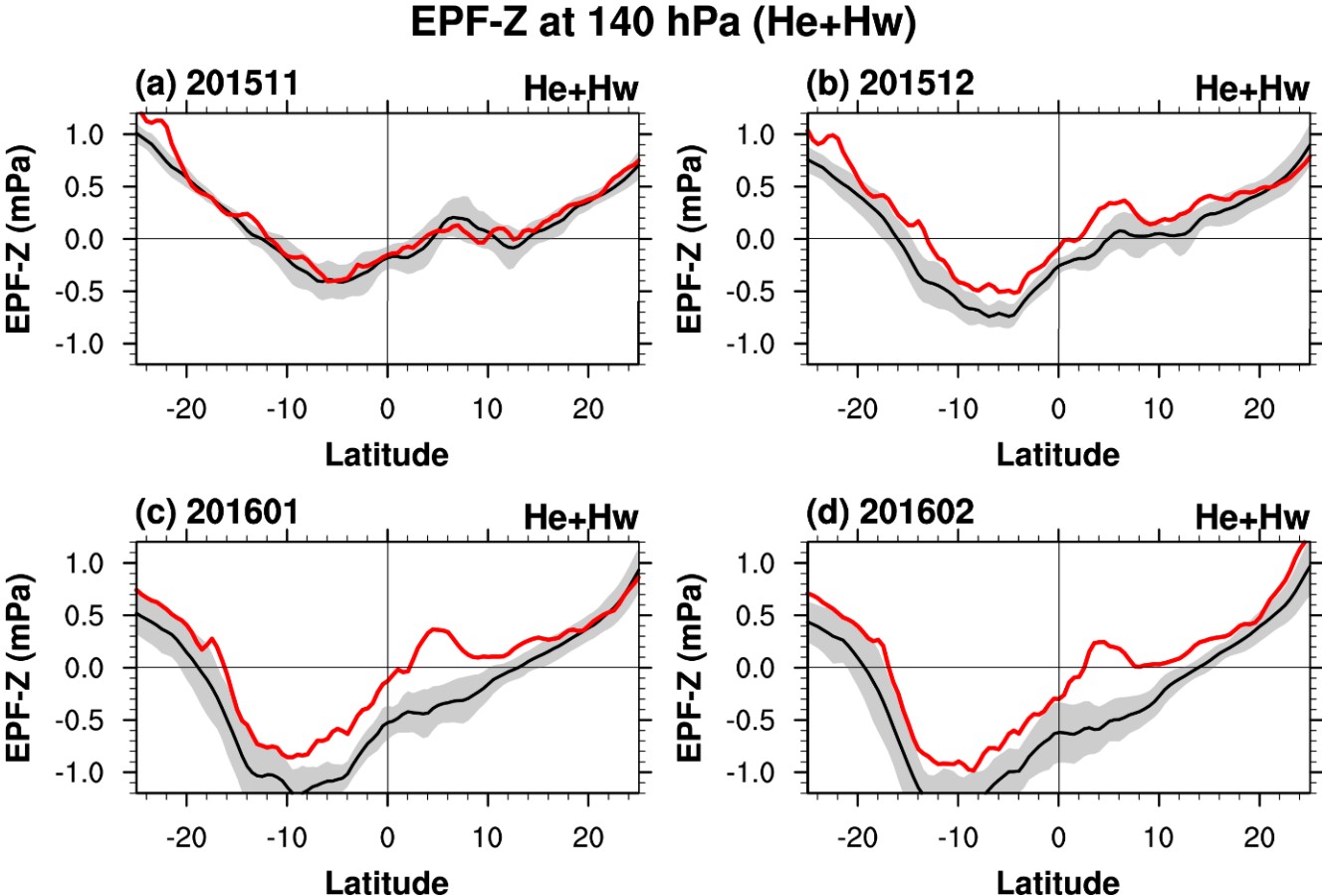

**Figure 12.** Vertical EP flux at 140 hPa for the $H_e + H_w$ waves [(i) $|k| > 20$ and $\omega > 0$ cpd or (ii) $|k| \leq 20$ and $\omega > 0.4$ cpd; approximately for the IG waves] in (a) November 2015, (b) December 2015, (c) January 2016, and (d) February 2016 (red)

and their monthly climatology (black) along with $\pm 1$ standard deviation (gray shading).




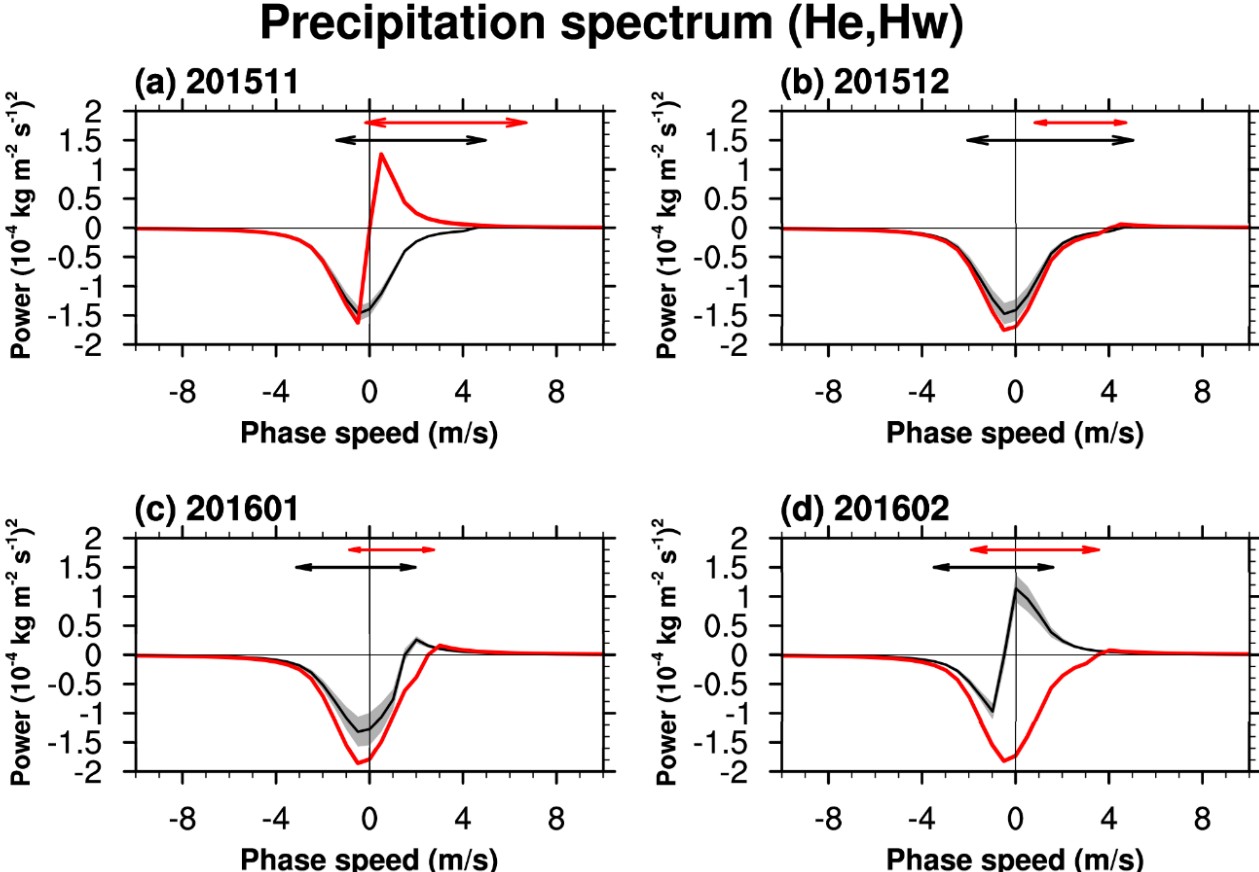

**Figure 13.** Phase-speed spectra of the precipitation in the spectral range of the $H_e + H_w$ waves [(i) $|k| > 20$ and $\omega > 0$ cpd or (ii) $|k| \leq 20$ and $\omega > 0.4$ cpd; approximately for the IG waves] averaged over 10°N–10°S in (a) November 2015, (b) December 2015, (c) January 2016, and (d) February 2016, and their monthly climatology (black) along with ±1 standard

deviation (gray shading). Note that the power is multiplied by a negative sign when the phase speed is smaller than the zonal-mean zonal wind at 140 hPa (i.e., source level). Double-sided arrows represent zonal wind ranges from 140 hPa to 70 hPa for the QBO disruption period (red) and the climatology (black).


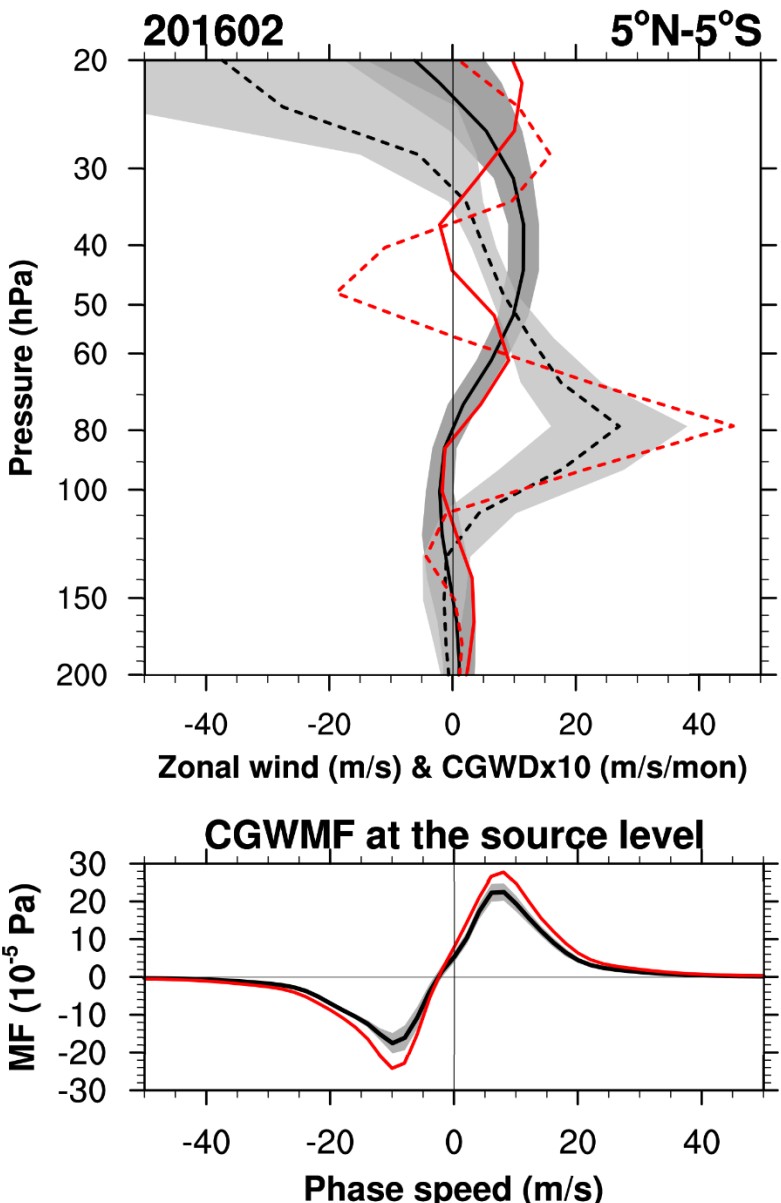

**Figure 14.** (Top) The zonal-mean zonal wind profile and the zonal-mean CGWD profile averaged over 5°N–5°S in February 2016 (red solid and red dashed, respectively) and those for the climatology (black solid and black dashed, respectively) with ± 1-standard deviation (dark-gray and light-gray shading, respectively). (Bottom) Phase-speed spectra of the zonal-mean zonal CGW momentum flux at the source level averaged over 5°N–5°S in February 2016 (red) and the climatology (black) with ± 1-standard deviation (gray shading).






**Figure 15.** Latitudinal distributions of (left) zonal-mean convective source spectrum and (right) wave-filtering and resonance
factor (WFRF) spectrum in (top) February 2016 and (bottom) the climatology. White and gray dashed lines in the convective

source spectrum denote zonal-mean zonal wind ($U$) and the moving-speed of convection ($c_{qh}$), respectively.



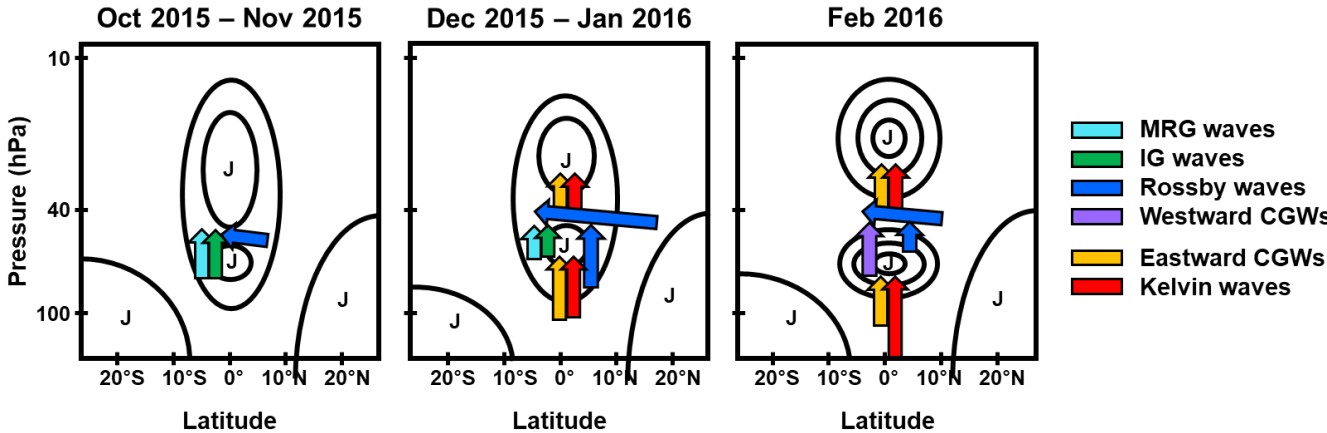

**Figure 16.** Schematic of the zonal wind evolution (black contour) and the anomalous wave forcing (arrow) during the QBO disruption in October

2015–November 2015 (left), December 2015–January 2016 (middle), and February 2016 (right). The "J" denotes a westerly jet.



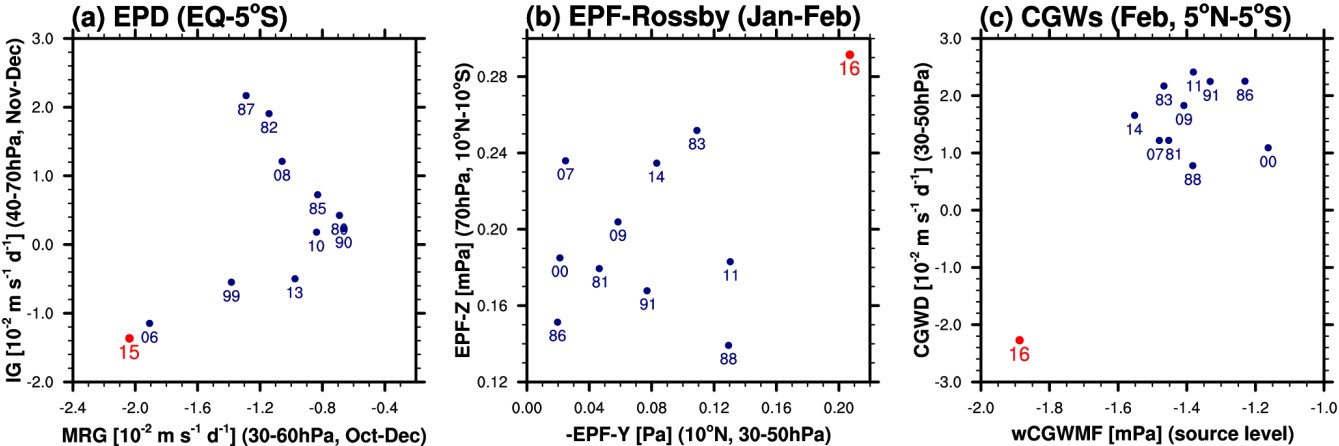

**Figure 17.** Scatter plots of the (a) EP flux divergence (EPD) for the MRG waves (*x*-axis) and that for the IG waves (*y*-axis) at 0°–5°S, averaged for October–December in 40–60 hPa and November–December in 30–70 hPa, respectively, (b) meridional EPF (multiplied by -1) at 10°N and 30–50 hPa (*x*-axis) and vertical EPF at 70 hPa in 10°N–10°S (*y*-axis) for the Rossby waves averaged for January–February, and (c) westward CGW momentum flux at the source level (*x*-axis) and the zonal-mean zonal CGWD (*y*-axis) in 30–50 hPa at 5°N–5°S averaged for February. Red dots denote the disruption year (2015–16) and dark-blue dots denote the other years with WQBO phases.