# Peer review of "Role of equatorial waves and convective gravity waves in the 2015–16 quasi-biennial oscillation disruption"

_Atmospheric Chemistry and Physics, 2020_

## Referee Comment (RC1) · Anonymous Referee #1 · 3 Sep 2020

This paper provides quite many missing pieces of the 2016 QBO disruption puzzle. Previous literature has concentrated on the role of equatorward-propagating extratropical waves, with only a couple papers giving equatorial wave modes any focus.

Overall, the effort made by the authors is quite impressive. All equatorial wave modes are separated and their effect on the wind structure is studied in detail, and additionally parameterized convective GWs are included in the study. As the authors mention, of course the CGW parameterization has some limitations (which parameterization doesn't?), but I think the approach is physically consistent and the CGW results are, at least, qualitatively correct.

[Figure]

The paper is a bit lengthy, but short summaries are provided for the trickier figures, which I appreciated. Overall well written, figures are mostly good, a nice sketch in Fig. 16 to put all together, interesting discussions and definitely worthy of publication in ACP. I only have a fair amount of minor comments, mainly to make some figures or methods easier to interpret for the reader, and in some parts to request a more detailed comparison with recent QBO disruption literature.

%%% Title: strictly speaking, it's not only planetary wavenumbers that you study, I suggest something more accurate "Role of equatorial waves and convective gravity-waves in the 2015/16 QBO disruption" or similar (it's ok to use the QBO abbreviation in the title, since all your potential audience will know what it means)

%%% p.2, l.31-33: specify that this is for extratropical latitudes when you refer to polar vortex and its downward impact.

%%% p.2, l.55: 'is the prerequisite for'

I think a better wording would be '... eq. waves preconditioned the extratropical Rossby wave breaking, ...'. prerequisite implies that without the MRG, the ex. Rossby wave breaking and QBO disruption wouldn't have happened, which to my knowledge cannot be assured 100%.

%%% p.2, l.56: '... each equatorial wave mode to the QBO...'

%%% p.3, l.87-88: sentence is repetitive, can be removed

%%% p.3, l.89-93: specify subsections where each item is done, e.g. specific wave types within section 3

%%% p.6, l.178: include the GPM dataset into section 2.1

%%% p.6, l.179: Here or perhaps within the supplement, justify why the magnitude and scaling are not crucial for both datasets in Fig. S1 to match: not the exact magnitude, but rather the shape of the vertical profiles is what's important, correct?

%%% p.6, l.180: SPB –> Since this abbreviation is not used later in the manuscript, I recommend to keep the full name, and remove the abbreviation from the first time it's used

%%% Figure 1: I suggest making 1(a) the lat.-height sections and 1(b) the vertical profiles and refer to them accordingly in the text.

Also, make it clear in the caption that the climatology is for WQBO years.

Y-axis labels: for better visibility I'd keep the pressure levels only, since the numbers with height in km get mixed up in between panels.

%%% p.7, l.188: In section 2.1 you say it will be referred to as WQBO climatology, you may want to rephrase that for consistency. Here, remind the reader that when you're talking about climatology, it refers to WQBO phase.

%%% p.7, l.199-200: '... positive (negative) drag on the zonal wind in regions of positive (negative) shear, ...'

%%% p.7, l.212: February –> March

%%% p.7, l.195-213: I miss some linking of these results with recent literature in this paragraph:

–> How do these results compare to Lin et al. (2019)? Especially for MRG, IG and CGW, it should be highlighted your results build upon those by Lin et al. (2019) (their Fig. 3 is quite similar to yours), that focus mostly on MRG in their discussions. Your eq. wave differentiation is more detailed, which is a big plus.

–> Regarding your Kelvin wave forcing results, I find it really neat that it helps maintain the two westerly jets. This was suggested/hinted by Li et al. (2020), who studied the long-lasting westerly jet around 20hPa, but your results really confirm this is the case. Li et al. (2020) also showed above-average upper-tropospheric Kelvin wave activity linked to El Nino. This should be mentioned/discussed within this paragraph.

–> Also, this is a very long paragraph, could be easily split into 2-3.

%%% Fig. 3: I suggest to move the climatology plot to the supplement (as it doesn't have crucial information for the main text), and make the rest bigger. Right now this figure is too busy and it's easy to get lost within, specially with the duplicate letters for the sub-panels.

%%% p.9, l.253-254: Mention to Lin et al. (2019) needed here. Your results are in line with theirs, plus the addition of IG in the preconditioning.

%%% p.10, l.294: origination –> just 'origin'

%%% p.11, l.324: I suggest making this (dividing by density) for all figures: the vertical/horizontal arrows without scaling for pressure in the previous plots, may lead the reader to underestimate upper level wave propagation at first glance, specially if one is quickly comparing upper tropospheric and stratospheric levels without paying much attention to the legend.

However, this is not a must: the figures have the divergence in pressure-independent units, and the units of the arrows are clearly shown - more than enough to correctly interpret everything with basic knowledge about EP flux.

%%% p.12, l.369-374: This belongs in the methods section

%%% Fig. 8(c-d): I suggest to, apart from making the c-d plot smaller, put the timeseries next to each other, or even merge them into one continuous timeseries.

%%% p.13, l.379-380:You may easily add a climatology line into Fig. 8(c-d)

%%% p.13, l.381: Could the authors perhaps provide a daily timeseries of MRG EPD (in the boxed region) to compare with with the q timeseries to assess this as a source? Would be a nice addition as Fig. 8(e-f)

%%% Fig. 9: I suggest to translate the y-axis unit (mean rain rates) into something more relatable, e.g. mm/day

%%% p.14, l.417-419: This was linked by Li et al. (2020) to the El Nino event that winter. Perhaps it would be useful to add a discussion somewhere in this section, about the overall increase in eq. wave activity and precipitation together with El Nino. Also a mention to Barton and McCormack (2017) could be added.

%%% p.14, l.440-442: There is a lot of new information and little justification about the source level here, I suggest you detail more about this in the methods section, and refer to the corresponding methods section when you start with this figure.

%%% p.15, l.446: A non-expert will need more explanation about the source level to understand the attribution made in this sentence. Again, this could be already detailed in the methods section and referred to here.

%%% Figure 13 and p.15, l. 447-460: I don't doubt the validity of your results and conclusions regarding Fig. 13, but it would be much easier to interpret if you showed the same plots for He and Hw separately. Mixing both makes this figure a bit confusing. Separating He and Hw will allow the reader to identify which fraction deviates more from climatology (and gets filtered by the wind shear) in a more straightforward way.

%%% p.15, l.462-467: You discuss the Kawatani paper in the next paragraph. Barton and McCormack (2017) showed important ENSO influence on the background winds and momentum fluxes below 30hPa (see e.g. their plots 3 and 4). It would be worth to add it into your discussion.

%%% p.16, l.488-490: Please detail a bit more what convective source and WFRF mean for the non-expert. Convective source spectrum is related to the movement of convection itself, WFRF to the GWs emmited from it, correct?

%%% p.16, l.496: higher static stability at which height range? Intuitively, deeper convection is related to tropospheric instability.

%%% p.16, l.497: please use relative terms: warmer / colder

%%% p.16, l.497-499: Rephrase this sentence to make it clear that El Nino increases

overall amount of convection in the tropics. There must be earlier studies (probably mentioned in the Domeisen review paper) providing this relation to say this with more certainty than 'possibly triggered by El Nino'

---

## Referee Comment (RC2) · Anonymous Referee #2 · 10 Sep 2020

This paper gives a comprehensive overview of the different wave forcings that act during the 2016 QBO disruption. The paper discusses the effect of wave forcing by extratropical Rossby waves, equatorial waves, and small scale convective gravity waves, and is therefore the most complete description of the QBO disruption so far. The paper is very well written and fits well into the scope of ACP. The paper is recommended for publication after addressing my minor comments as detailed below.

Main comments:

(1) For the spectral analysis time segments of 90 days are used after applying sine and cosine windows at the first and last 30 days. According to Parseval's theorem, this will

lead to an underestimation of spectral amplitudes, and, on average, EPF and EPD will be ∼30% underestimated. This relatively small effect will not affect the basic results of the paper, but it should be mentioned.

(2) I am not sure whether the preconditioning of the QBO disruption by MRG EPD and IG EPD in October 2015 is a reliable result. Firstly, the magnitude of EPD is quite small, secondly, the EPD in October 2015 looks different in ERA-Interim, as can be seen in the supplement.

Specific comments:

p2, l33: You may want to include more recent work on the effect of the QBO on surface weather and climate, for example Kidston et al. (2015), or Gray et al. (2018).

Kidston, J., A. A. Scaife, S. C. Hardiman, D. M. Mitchell, N. Butchart, M. P. Baldwin, and L. J. Gray (2015), Stratospheric influence on tropospheric jet streams, storm tracks and surface weather, Nat. Geosci., 8, 433-440.

Gray, L. J., J. A. Anstey, Y. Kawatani, H. Lu, S. Osprey, and V. Schenzinger (2018), Surface impacts of the Quasi Biennial Oscillation, Atmos. Chem. Phys., 18, 8227-8247.

p3, l73: The reference Evan et al., JAS, 2012 does not fit here. Evan et al., JAS, 2012 discuss intermediate-scale tropical inertia-gravity waves of horizontal wavelengths in the 5000km range, and not the effect of small scale gravity waves.

This reference should be replaced by Evan et al., JGR, 2012 which is a WRF simulation of the QBO forcing by small scale gravity waves of horizontal wavelengths >270km. In addition to the listed model studies you should also include observational evidence of the effect of small scale gravity waves, for example Ern et al., JGR, 2014.

Evan, S., M. J. Alexander, and J. Dudhia (2012), WRF simulations of convectively generated gravity waves in opposite QBO phases, J. Geophys. Res., 117, D12117, doi:10.1029/2011JD017302.

Ern, M., F. Ploeger, P. Preusse, J. C. Gille, L. J. Gray, S. Kalisch, M. G. Mlynczak, J. M. Russell III, and M. Riese (2014), Interaction of gravity waves with the QBO: A satellite perspective, J. Geophys. Res. Atmos., 119, 2329-2355, doi:10.1002/2013JD020731.

p3, l88: "All" gravity waves is not correct! As stated in l.82/83, this paper discusses only small scale gravity waves of horizontal wavelengths <100-200km. However, there is also considerable QBO forcing by small scale gravity waves of horizontal wavelengths >200km, as can be seen from Evan et al., JGR, 2012.

p4, l106: On the selection of WQBO cases: Please state more clearly that the definition of the WQBO as used in this paper focuses on QBO situations that are comparable to that of the 2016 QBO disruption. Generally, there should be WQBO periods or WQBO onsets also in other months.

p5, l134/135: As EPF is obtained by summing in the spectral space (l.142), applying cosine windows will lead to an underestimation of spectral amplitudes, and also of EPF, and EPD. See Main Comment (1).

p9, l253/254: I am not sure whether it is a reliable result that IG and MRG would act as a preconditioning at 40hPa in October 2015 before Rossby waves can take effect! Please note that Fig.4b (for MERRA2) and Fig.S5 in the supplement (for ERA-Interim) look quite different! The preconditioning effect that you suggest seems to be much weaker than the differences between the two reanalyses.

p10, l281-285: Another reason for this difference could be the part of the gravity wave spectrum that is neither covered by the CGW scheme, nor resolved or parameterized in MERRA2. It should be emphasized that MERRA2 is not a free-running model! There will be model imbalances that are caused by data assimilation. Data assimilation can therefore correct misrepresentations of the gravity wave forcing by the nonorographic GWD parameterization.

Fig.6: Here you identify source regions of Rossby waves by positive EPD coinciding

with upward directed EPF. At 15N around 150hPa EPF is directed downward coinciding with negative EPD. Do you think this is another source region of Rossby waves? As EPF is directed downward, it looks like these waves cannot propagate into the stratosphere.

p11, l327: that the positive EPD region is a source region of the westward-propagating waves. -> that the positive EPD region should be a source region of westward-and-upward propagating waves.

p12, l370: It should be mentioned that Eq.(3) includes both barotropic and baroclinic instability. Did you check which term is stronger - the barotropic term (meridional gradients), or the baroclinic term (vertical gradients)? Coy et al., 2017 claimed that barotropic instability would be stronger. In your Fig.8, this does not fit the title and the figure caption saying "baroclinic instability".

p17, l513: barotropic instability -> barotropic and/or baroclinic instability

Technical comments:

p5, l124: the parameter \hat{f} is not used in Eq.(1)

p6, l175: is lower than 700 hPa -> is at altitudes lower than 700 hPa

Fig.1: colorbar should be m/s, but is m/s/month

Fig.3: one of the colorbars should refer to the wind in m/s, but both colorbars give tendencies in m/s/month

p8, l222: (Fig. 3a) -> (Fig. 3c)

p9, l256/257: Figure S4 is the same figure with Fig. 3 but using ERA-I data. -> Figure S4 shows the same as Fig. 3 but using ERA-I data.

p9, l264: (Fig. 4c) -> (Fig. 4a, dotted line)

caption of Fig.7, l795: (c) January 2016, -> (c) December 2015,

caption of Fig.7, l799: where the EPD is smaller than -> where the EPD is stronger than

p13, l390: affects -> affect

caption of Fig.9, p35, l814: January 2016, and (b) -> January 2016, and (d)

---

## Referee Comment (RC3) · Anonymous Referee #3 · 18 Sep 2020

Summary: This paper provides a detailed examination of the equatorial wave structures and their evolution during October 2015 through February 2016, a time when the quasi-biennial oscillation experienced a significant disruption. This investigation, based on MERRA-2, a global assimilation system, breaks down the wind, temperature, and precipitation fields into Rossby, Mixed Rossby Gravity (MRG), inertia-Gravity (IG), and Convective Gravity (CG) waves. A novel aspect of this work is the use of a convective gravity wave parameterization to calculate the CG wave effects. The different Eliassen Palm (EP) fluxes and their divergences are evaluated. Quantities calculated for the QBO disruption are compared to their corresponding climate signatures. The results show how during October-November of 2015 MRG and IG waves acted to precondition

the QBO winds before the strong Rossby waves that occurred in 2016 and created the anomalous QBO easterlies. Why these waves were stronger than usual during this time is still unknown.

Strengths: This work provides a comprehensive view of the UT/LS equatorial waves during the QBO disruption. It expands on the work of Lin et al. (2019) by including aspects of the tropospheric forcing by precipitation and the addition of CG wave model and also differs in the choice of assimilation system from Lin et al. (2019). Figure 16 provides an especially useful summary of the changes roles played by the different waves.

Weaknesses: No major weaknesses. There are a few points that could be improved for clarity that are detailed below. There are a large number of figures. These provide a comprehensive record but it can be difficult for readers to locate the features of interest as described in the text.

Recommendation: Publish after minor revisions noted below. This is a well written, well organized, manuscript. The figures are appropriately captioned and the abstract accurately summaries the work. The topic should be of interest to many readers of ACP interested the the QBO and equatorial waves.

Comments:

Line 44: "El Nino" Coy et al. (2017) only mentioned a possibility of an ENSO connect. The terms El Nino or ENSO are not found in Osprey et al. (2016) so this sentence should be rewritten.

Line 58: Should be "Coy et al. (2017)". There is a Coy et al. (2016) describing the MERRA-2 QBO before the disruption that could probably be mentioned somewhere in the data section: Coy, L., K. Wargan, A. M. Molod, W. R. McCarty, and S. Pawson. 2016. "Structure and Dynamics of the Quasi-Biennial Oscillation in MERRA-2." J. Climate, 29:14: 5339-5354 [10.1175/jcli-d-15-0809.1]

[Figure]

Line 67: Each MERRA-2 data set has a DOI number that researchers are encouraged to reference and clarifies exactly what data set was used. For example from https://disc.gsfc.nasa.gov/datasets/M2I3NVASM_5.12.4/summary?keywords=%22MERRA-2%22 To cite the data in publications: Global Modeling and Assimilation Office (GMAO) (2015), MERRA-2 inst3_3d_asm_Nv: 3d,3-Hourly,Instantaneous,Model-Level,Assimilation,Assimilated Meteorological Fields V5.12.4, Greenbelt, MD, USA, Goddard Earth Sciences Data and Information Services Center (GES DISC), Accessed: [Data Access Date], 10.5067/WWQSXQ8IVFW8

Lines 248-249: How is the budget formulated so that resolved MRG and IG wave forcing acts to "enhance" the momentum budget residual, REQ? Maybe this could be rewritten for clarity.

Lines 277-287: This is a good discussion of the CGWD. It would help to see the vertical zonal mean zonal wind shear at 40 hPa as a part of Fig. 4 as that should determined in large part the GWD forcing. In addition the meridional shear of the zonal mean zonal wind across the equator might correspond to the changes in the Rossby wave forcing and be helpful to see plotted.

Lines 357-362: Figures 3, 4, and 7 all illustrate aspects of MRG waves, however it is difficult to put together a consistent picture of support for the mid-jet easterly acceleration described here. Most of the easterly acceleration appears to take place in the regions of strong wind shear, not mid-jet. In particular the contribution from MRG waves in Fig. 4 at 40 hPa is small and appears nearly constant in time. The mid-jet should be identified more quantitatively and MRG wave aspects calculated with respect to the jet at each month, at least for the earlier months, to justify this conclusion.

Lines 375-392: This instability analysis is based on zonal mean winds. A stronger case for instability might be possible with non-zonally average winds, especially when the focus is on the relatively small region defined by the box in Figure 8.

Lines 443-485: The "...apparent positive wind shear..." Is difficult to find in Fig. 14a.

The specific levels should be specified in the text.

Lines 499-502: The white and gray curves described in the text appear to be different from the Fig. 15 caption description.

Minor Comments:

Line 375: The units in Fig. 8 suggest that q_y is plotted not q_phi. This is a small point. Perhaps the units could be described in the figure caption.

Lines 732-733: The year is missing from the reference.

---

## Referee Comment (RC4) · Anonymous Referee #4 · 18 Sep 2020

Review of "Role of equatorial planetary and gravity waves in the 2015–16 quasi-biennial oscillation disruption" by Kang et al.

Recommendation: accept after very minor revision

General comments

The authors investigated the relative contribution of each resolved wave and parameterized waves to the QBO disruption in 2015-2016. They have shown that MRG and westward IG weakened the QBO and then led to extratropical Rossby breaking at the QBO jet core at 40 hPa. They also investigated the roles of CGWs obtained from an offline CGW parameterization that author's group has developed and showed the importance of variable wave sources. There have been several studies to investigate the mechanism of the 2015-2016 QBO disruption. I think this paper is the most comprehensive study among them. I believe this paper is suitable for the publication in ACP. My recommendation is published after very minor revision. I have a few comments added below.

(1) MRG are confined to the range |k|<=20 and omega <0.75 cpd in the symmetric spectrum. I think zonal |k|<=20 is a little wide for the MRG. Presumably |k|<=~10 would be better. Westward IGWs should be included in this definition. How much do the results depend on the ranges of |k|? I guess the relative contribution of MRG, shown in Table 1 and Figure 4, would be changed. One good point to answer this concern is to mention the dominant zonal wavenumber ranges for the MRG to force the QBO. I guess 3 < k < 6, but am not sure. I would suggest authors at least to mention the dependence of |k| selection to the quantitative results.

(2) L216: "The required wave forcing term (REQ) is calculated as a residual by subtracting the advection terms from the zonal wind tendency in the TEM equation"

When calculating REQ, do the authors consider the first term on the left of Eq. (1), that is meridional advection term, which is normally very small near the equator? In my experiences, the meridional advection term has also some values off the equator even at ~5 degrees, which cannot be sometimes negligible.

(3) L258: "although the magnitudes of the REQ and wave forcing (vertical advection) in ERA-I is generally stronger (weaker) than that in MERRA-2"

I guess one possible reason for this is the different values of w* between MERRA-2 and ERA-I. As you know, the representation of BDC is quite different quantitatively among reanalyses as the S-RIP project has indicated.

(4) I would suggest to refer the paper by Dunkerton (2016, GRL, https://doi.org/10.1002/2016GL070921). Dunkerton's paper, published just after

the QBO disruption, discussed some presumable mechanisms, which would be now useful for the current study.

(5) Figure 4(c): The explanation lines of Rossby-Y & Rossby-Z are hard to see. Please expand the lines.

---

## Author Comment (AC1) · 13 Oct 2020

General Comment:

This paper provides quite many missing pieces of the 2016 QBO disruption puzzle. Previous literature has concentrated on the role of equatorward-propagating extratropical waves, with only a couple papers giving equatorial wave modes any focus. Overall, the effort made by the authors is quite impressive. All equatorial wave modes are separated and their effect on the wind structure is studied in detail, and additionally parameterized convective GWs are included in the study. As the authors mention, of course the CGW parameterization has some limitations (which parameterization doesn't?),

but I think the approach is physically consistent and the CGW results are, at least, qualitatively correct. The paper is a bit lengthy, but short summaries are provided for the trickier figures, which I appreciated. Overall well written, figures are mostly good, a nice sketch in Fig. 16 to put all together, interesting discussions and definitely worthy of publication in ACP. I only have a fair amount of minor comments, mainly to make some figures or methods easier to interpret for the reader, and in some parts to request a more detailed comparison with recent QBO disruption literature.

Response: First of all, we would like to thank the reviewer for the time spent on reviewing our long manuscript. All reviews have been beneficial and made us aware of important points which had to be addressed. We, the authors, are therefore thankful for the reviewer's contribution to improve the manuscript. We carefully addressed all comments and tried our best to improve the manuscript based on the suggestions and comments.

Specific Comments:

1. Title: strictly speaking, it's not only planetary wavenumbers that you study, I suggest something more accurate "Role of equatorial waves and convective gravity waves in the 2015/16 QBO disruption" or similar (it's ok to use the QBO abbreviation in the title, since all your potential audience will know what it means)

Response: Thank you for your good suggestion. Following the suggestion, the title is changed to "Role of equatorial waves and convective gravity waves in the 2015–16 quasi-biennial oscillation disruption". [p.1, L1–2]

2. p.2, l.31-33: specify that this is for extratropical latitudes when you refer to polar vortex and its downward impact.

Response: It is specified as suggested. [p.2, L32–33]

3. p.2, l.55: 'is the prerequisite for' I think a better wording would be '... eq. waves preconditioned the extratropical Rossby wave breaking, ...'. prerequisite implies that

without the MRG, the ex. Rossby wave breaking and QBO disruption wouldn't have happened, which to my knowledge cannot be assured 100%.

Response: We agree with you. The sentence is modified as suggested. [p.2, L56]

4. p.2, l.56: '... each equatorial wave mode to the QBO...'

Response: It is changed as suggested. [p.2, L57]

5. p.3, l.87-88: sentence is repetitive, can be removed

Response: The sentence is removed as suggested.

6. p.3, l.89-93: specify subsections where each item is done, e.g. specific wave types within section 3

Response: Thank you for your good suggestion. We specified subsections where each item is done. [p. 3, L90–93]

7. p.6, l.178: include the GPM dataset into section 2.1

Response: Thank you for pointing this out. We included the GPM dataset in Sect. 2.1. [p. 4, L106–108]

8. p.6, l.179: Here or perhaps within the supplement, justify why the magnitude and scaling are not crucial for both datasets in Fig. S1 to match: not the exact magnitude, but rather the shape of the vertical profiles is what's important, correct?

Response: Thank you for your comment. The spatiotemporal variations in the magnitude of convective heating rate are important, but the overall magnitude largely depends on the resolution of the data provided. Therefore, the 'exact' magnitude would be less important compared to the shape of the heating profiles. The difference in the overall magnitude can be adjusted by the conversion factor (Kang et al., 2017), a scale factor that constitutes convective source spectrum, when calculating convective gravity wave momentum flux. The related sentence is included in the revised supplement.

[Fig. S2]

9.  p.6, l.180: SPB –> Since this abbreviation is not used later in the manuscript, I recommend to keep the full name, and remove the abbreviation from the first time it's used

Response: It is modified as suggested. [p.3, L85; p.7, L186]

10. Figure 1: I suggest making 1(a) the lat.-height sections and 1(b) the vertical profiles and refer to them accordingly in the text. Also, make it clear in the caption that the climatology is for WQBO years. Y-axis labels: for better visibility I'd keep the pressure levels only, since the numbers with height in km get mixed up in between panels.

Response: Thank you for your good suggestion. Following the suggestion, we marked (a) and (b) in Fig. 1 and deleted labels of the height axis. We also clarified the meaning of the climatology in the figure caption of Fig. 1 in the revised manuscript. [Fig. 1]

11. p.7, l.188: In section 2.1 you say it will be referred to as WQBO climatology, you may want to rephrase that for consistency. Here, remind the reader that when you're talking about climatology, it refers to WQBO phase.

Response: Thank you for pointing this out! We revised the statement in Sect. 2.1 that the 10-selected years will be referred to as the 'climatology' for conciseness. We also remind the reader of this in the results section. [p.4, L114; p.7, L205–206]

12. p.7, l.199-200: '... positive (negative) drag on the zonal wind in regions of positive (negative) shear, ...'

Response: It is changed as suggested. [p.8, L217–218]

13. p.7, l.212: February –> March

Response: Thank you! It is corrected. [p.8, L234]

14.  p.7, l.195-213: I miss some linking of these results with recent literature in this

paragraph: –> How do these results compare to Lin et al. (2019)? Especially for MRG, IG and CGW, it should be highlighted your results build upon those by Lin et al. (2019) (their Fig. 3 is quite similar to yours), that focus mostly on MRG in their discussions. Your eq. wave differentiation is more detailed, which is a big plus. –> Regarding your Kelvin wave forcing results, I find it really neat that it helps maintain the two westerly jets. This was suggested/hinted by Li et al. (2020), who studied the long-lasting westerly jet around 20hPa, but your results really confirm this is the case. Li et al. (2020) also showed above-average upper-tropospheric Kelvin wave activity linked to El Nino. This should be mentioned/discussed within this paragraph. –> Also, this is a very long paragraph, could be easily split into 2-3.

Response: Thank you for your detailed comments. We modified the paragraph as suggested: - We included a comparison between Fig. 2 and Lin et al. (2019)'s Fig. 3. We also emphasized that the contribution to the QBO disruption from the IG waves is first shown in this study. - We included a discussion on the long-lasting jet near 20 hPa and its contribution from Kelvin waves, referring to Li et al. (2020). - We split a paragraph into three. [p. 8, L221–235]

15. Fig. 3: I suggest to move the climatology plot to the supplement (as it doesn't have crucial information for the main text), and make the rest bigger. Right now this figure is too busy and it's easy to get lost within, specially with the duplicate letters for the sub-panels.

Response: Thank you for your comment. Although we respect the reviewer's suggestion, we decided to keep the figure in the revised manuscript, as two figures (disruption and climatology) together can clearly show how the structures of the wind and wave forcings are unusual during the disruption. We instead changed letters for sub-panels and adjusted each figure to increase visibility as much as possible. [Fig. 3]

16. p.9, l.253-254: Mention to Lin et al. (2019) needed here. Your results are in line with theirs, plus the addition of IG in the preconditioning.

Response: We mentioned Lin et al. (2019) in the revised manuscript as suggested. [p. 10, L279–280]

17. p.10, l.294: origination –> just 'origin'

Response: It is corrected. [p.11, L325]

18. p.11, l.324: I suggest making this (dividing by density) for all figures: the vertical/ horizontal arrows without scaling for pressure in the previous plots, may lead the reader to underestimate upper level wave propagation at first glance, specially if one is quickly comparing upper tropospheric and stratospheric levels without paying much attention to the legend. However, this is not a must: the figures have the divergence in pressure-independent units, and the units of the arrows are clearly shown - more than enough to correctly interpret everything with basic knowledge about EP flux.

Response: Thank you for your comment. Following the reviewer's suggestion, we made all figures divided by air density except for Fig. 2, because the EPF for the CGWs in Fig. 2 divided by air density was too strong in the upper stratosphere. [Figs. 5, 7, and 11]

19. p.12, l.369-374: This belongs in the methods section

Response: We moved the statements to the methods section (Sect. 2.5). [p. 7, L190–198]

20. Fig. 8(c-d): I suggest to, apart from making the c-d plot smaller, put the timeseries next to each other, or even merge them into one continuous timeseries.

Response: Thank you for your suggestion. We merged Figs. 8c and 8d into one continuous timeseries. [Fig. 8]

21. p.13, l.379-380: You may easily add a climatology line into Fig. 8(c-d)

Response: Following the reviewer's suggestion, we added a climatology line with $\pm 1$ standard deviation range in Fig. 8c. [Fig. 8]

22. p.13, l.381: Could the authors perhaps provide a daily timeseries of MRG EPD (in the boxed region) to compare with the q timeseries to assess this as a source? Would be a nice addition as Fig. 8(e-f)

Response: Thank you for the good suggestion. Following the reviewer's suggestion, we added the daily EPD time series for the MRG waves as a new figure Fig. 8d, and included a discussion that the magnitude of the positive EPD increases as the number of grids with q_ÏŢ<0 increases. [Fig. 8; p. 13, L406–407]

23. Fig. 9: I suggest to translate the y-axis unit (mean rain rates) into something more relatable, e.g. mm/day

Response: Thank you for the good point. The unit is changed to "mm/day" in the revised manuscript. [Fig. 9]

24. p.14, l.417-419: This was linked by Li et al. (2020) to the El Nino event that winter. Perhaps it would be useful to add a discussion somewhere in this section, about the overall increase in eq. wave activity and precipitation together with El Nino. Also a mention to Barton and McCormack (2017) could be added.

Response: Thank you for your suggestion! We included a discussion on the enhanced Kelvin wave activity related to the El Niño event and the overall enhancement in the equatorial wave activity by referring to Li et al. (2020). We did not include a paper by Barton and McCormack (2017) here because as far as we know, their focus was not on the equatorial waves but on the midlatitude Rossby waves and their equatorward propagation. [p.15, L447–449]

25. p.14, l.440-442: There is a lot of new information and little justification about the source level here, I suggest you detail more about this in the methods section, and refer to the corresponding methods section when you start with this figure.

Response: Following the reviewer's suggestion, we included the details about the source level in the method section (Sect. 2.3) by including a new figure (Figure S1)

and refer to them when we started to describe Fig. 12 in the revised manuscript. The changes in the direction of the vertical EPF and the sign of the EPD appear at 150 hPa and below (140 hPa and above) for H_e (H_w) waves. Our focus is on the westward waves, so we simply assumed the source level of H_e+H_w as 140 hPa. [p.6, L160–162; p.15, L471]

26.  p.15, l.446: A non-expert will need more explanation about the source level to understand the attribution made in this sentence. Again, this could be already detailed in the methods section and referred to here.

Response: As discussed in the Comment #25, we included more explanation in the methods section. [p.6, L160–162; p.16, L475]

27.  Figure 13 and p.15, l.  447-460: I don't doubt the validity of your results and conclusions regarding Fig. 13, but it would be much easier to interpret if you showed the same plots for He and Hw separately. Mixing both makes this figure a bit confusing. Separating He and Hw will allow the reader to identify which fraction deviates more from climatology (and gets filtered by the wind shear) in a more straightforward way.

Response: We plotted Fig. 13 mimicking the phase-speed spectrum of the GW parameterization (e.g., Fig. 1 of Beres et al. 2004; Fig. 2 of Kang et al., 2017) which, together with wind profiles, can well represent the sign and magnitude of the potential GWD. Therefore, we would like to keep Fig. 13 as the original. We understand that some readers might be confused and there could be an offset between eastward and westward waves, so we provided the same figure as Fig. 13 but for the eastward and westward propagating parts separately, in the revised supplement (Figure S9). Please note that the figure is not separated into H_e and H_w but into the eastward and westward waves. This is because we defined H_e where k>0 and H_w where k<0 based on the assumption that the zonal wind is close to zero near the equator, but precisely, the direction of the waves is determined by the wind speed at the altitude where the waves are generated (i.e., source level). Since we assumed the source level as 140

hPa (please refer to Response #25), the spectrum where the phase speed is greater (smaller) than zonal wind at 140 hPa is defined as eastward (westward) waves. [p.16, L492–493; Fig. S9]

28. p.15, l.462-467: You discuss the Kawatani paper in the next paragraph. Barton and McCormack (2017) showed important ENSO influence on the background winds and momentum fluxes below 30hPa (see e.g. their plots 3 and 4). It would be worth to add it into your discussion.

Response: Following the reviewer's suggestion, we mentioned ENSO influence on the background winds in the revised manuscript referring to Barton and McCormack (2017). However, ENSO influence on the momentum fluxes below 30 hPa in Fig. 4 of Barton and McCormack (2017) is shown for the horizontal component only, while IG waves have dominant vertical component. Therefore, no further discussions regarding to the work by Barton and McCormack (2017) are made. [p.16, L502–504]

29. p.16, l.488-490: Please detail a bit more what convective source and WFRF mean for the non-expert. Convective source spectrum is related to the movement of convection itself, WFRF to the GWs emitted from it, correct?

Response: Yes, it is correct. The convective source spectrum is about the size, magnitude, and moving speed of the convection and the WFRF is about the shape of the GW spectra emitted from the convection, which is related to the vertical configuration of the convective heating, the critical-level filtering, and resonance between the forced mode and natural mode. The combined effect of the two determines the magnitude and spectral shape of the CGW momentum flux at the source level. We included more explanation in the revised manuscript, as suggested. [p.17, L522–526]

30. p.16, l.496: higher static stability at which height range? Intuitively, deeper convection is related to tropospheric instability.

Response: Thank you for your comment! We found that the original statement was

somewhat unclear. We included the height range, 200–300 hPa, and modified the sentence more clearly referring to He et al. (2019) in the revised manuscript. According to He et al. (2019), high tropical static stability appears under global warming because of the enhanced surface temperature resulting in a smaller moist adiabatic temperature lapse rate. Please note that 2016 is the warmest year on record for the global-mean surface temperature (GISTEMP Team, 2020). [p.17, L531–533]

31. p.16, l.497: please use relative terms: warmer / colder

Response: Thank you for your suggestion, but the related sentence was removed during the revision process.

32. p.16, l.497-499: Rephrase this sentence to make it clear that El Nino increases overall amount of convection in the tropics. There must be earlier studies (probably mentioned in the Domeisen review paper) providing this relation to say this with more certainty than 'possibly triggered by El Nino'

Response: Thank you for your suggestion. We rephrased the sentence that El Niño increases the overall amount of convection in the tropics referring to Geller et al. (2016) and Kawatani et al. (2019). [p.17, L529–531]

References

Beres, J. J., M. J. Alexander, and J. R. Holton: A method of specifying the gravity wave spectrum above convection based on latent heating properties and background wind. J. Atmos. Sci., 61, 324–337, doi:10.1175/1520-0469(2004)061<0324:AMOSTG>2.0.CO;2, 2004.

Geller, M. A., T. Zhou, and W. Yuan: The QBO, gravity waves forced by tropical convection, and ENSO, J. Geophys. Res. Atmos., 121, 8886–8895, doi:10.1002/2015JD024125, 2016.

GISTEMP Team: GISS Surface Temperature Analysis (GISTEMP), version 4. NASA Goddard Institute for Space Studies. Dataset accessed 2020-09-10 at

https://data.giss.nasa.gov/gistemp/, 2020.

He, C., Y. Wang, and T. Li: Weakened impact of the developing El Niño on tropical Indian Ocean climate variability under global warming, J. Climate, 32, 7265–7279, https://doi.org/10.1175/JCLI-D-19-0165.1, 2019.

Please also note the supplement to this comment:
https://acp.copernicus.org/preprints/acp-2020-791/acp-2020-791-AC1-supplement.pdf

**Supplement:**

**Response to Reviewers' Comments**

**Min-Jee Kang, Hye-Yeong Chun, and Rolando R. Garcia**

**October 13, 2020**

Dear editor and reviewers,

We received four reviews for our manuscript "Role of equatorial planetary and gravity waves in the 2015–16 quasi-biennial oscillation disruption". First of all, we would like to thank all the reviewers for their time spent on reviewing our long manuscript. All reviews have been beneficial and made us aware of important points which had to be addressed. We, the authors, are therefore thankful for their contribution to improve the manuscript. We carefully addressed all comments and tried our best to improve the manuscript based on the suggestions and comments.

During the revision process, we changed the title to "Role of equatorial waves and convective gravity waves in the 2015–16 quasi-biennial oscillation disruption". In addition, we newly included (i) the time series of the meridional and vertical wind shears at 40 hPa in Fig. 4d, (ii) the climatology of the daily $\bar{q}_\phi$ in Fig. 8c, and (iii) the daily EPD for the MRG waves in Fig. 8d. Following the reviewers' suggestions, we added new supplementary figures (Figures S1, S6, and S9) and clarified some points that were unclear in the original manuscript.

We include a point-by-point response to each comment in the following paragraphs. We indicate the original comment of the respective reviewer in blue color and our answer in black color. In addition, we provide a tracked-changed version of the manuscript.

Sincerely,

Hye-Yeong Chun

**Response to Reviewer #1's Comments**

**General Comment:**

This paper provides quite many missing pieces of the 2016 QBO disruption puzzle. Previous literature has concentrated on the role of equatorward-propagating extratropical waves, with only a couple papers giving equatorial wave modes any focus.

Overall, the effort made by the authors is quite impressive. All equatorial wave modes are separated and their effect on the wind structure is studied in detail, and additionally parameterized convective GWs are included in the study. As the authors mention, of course the CGW parameterization has some limitations (which parameterization doesn't?), but I think the approach is physically consistent and the CGW results are, at least, qualitatively correct.

The paper is a bit lengthy, but short summaries are provided for the trickier figures, which I appreciated. Overall well written, figures are mostly good, a nice sketch in Fig. 16 to put all together, interesting discussions and definitely worthy of publication in ACP. I only have a fair amount of minor comments, mainly to make some figures or methods easier to interpret for the reader, and in some parts to request a more detailed comparison with recent QBO disruption literature.

**Specific Comments:**

1)  Title: strictly speaking, it's not only planetary wavenumbers that you study, I suggest something more accurate "Role of equatorial waves and convective gravity waves in the 2015/16 QBO disruption" or similar (it's ok to use the QBO abbreviation in the title, since all your potential audience will know what it means)

Thank you for your good suggestion. Following the suggestion, the title is changed to "Role of equatorial waves and convective gravity waves in the 2015–16 quasi-biennial oscillation disruption". [p.1, L1–2]

2)  p.2, l.31-33: specify that this is for extratropical latitudes when you refer to polar vortex and its downward impact.

It is specified as suggested. [p.2, L32–33]

3) p.2, l.55: 'is the prerequisite for'

I think a better wording would be '... eq. waves preconditioned the extratropical Rossby wave breaking, ...'. prerequisite implies that without the MRG, the ex. Rossby wave breaking and QBO disruption wouldn't have happened, which to my knowledge cannot be assured 100%.

We agree with you. The sentence is modified as suggested. [p.2, L56]

4) p.2, l.56: '... each equatorial wave mode to the QBO...'

It is changed as suggested. [p.2, L57]

5) p.3, l.87-88: sentence is repetitive, can be removed

The sentence is removed as suggested.

6) p.3, l.89-93: specify subsections where each item is done, e.g. specific wave types within section 3

Thank you for your good suggestion. We specified subsections where each item is done. [p. 3, L90–93]

7) p.6, l.178: include the GPM dataset into section 2.1

Thank you for pointing this out. We included the GPM dataset in Sect. 2.1. [p. 4, L106–108]

8) p.6, l.179: Here or perhaps within the supplement, justify why the magnitude and scaling are not crucial for both datasets in Fig. S1 to match: not the exact magnitude, but rather the shape of the vertical profiles is what's important, correct?

Thank you for your comment. The spatiotemporal variations in the magnitude of convective heating rate are important, but the overall magnitude largely depends on the resolution of the data provided. Therefore, the 'exact' magnitude would be less important compared to the shape of the heating profiles. The difference in the overall magnitude can be adjusted by the conversion factor (Kang et al., 2017), a scale factor that constitutes convective source spectrum, when calculating convective gravity wave momentum flux. The related sentence is included in the revised supplement. [Fig. S2]

9) p.6, l.180: SPB –> Since this abbreviation is not used later in the manuscript, I recommend

to keep the full name, and remove the abbreviation from the first time it's used
It is modified as suggested. [p.3, L85; p.7, L186]

10) Figure 1: I suggest making 1(a) the lat.-height sections and 1(b) the vertical profiles and refer to them accordingly in the text.
Also, make it clear in the caption that the climatology is for WQBO years.
Y-axis labels: for better visibility I'd keep the pressure levels only, since the numbers with height in km get mixed up in between panels.
Thank you for your good suggestion. Following the suggestion, we marked (a) and (b) in Fig. 1 and deleted labels of the height axis. We also clarified the meaning of the climatology in the figure caption of Fig. 1 in the revised manuscript. [Fig. 1]

11) p.7, l.188: In section 2.1 you say it will be referred to as WQBO climatology, you may want to rephrase that for consistency. Here, remind the reader that when you're talking about climatology, it refers to WQBO phase.
Thank you for pointing this out! We revised the statement in Sect. 2.1 that the 10-selected years will be referred to as the 'climatology' for conciseness. We also remind the reader of this in the results section. [p.4, L114; p.7, L205–206]

12) p.7, l.199-200: '... positive (negative) drag on the zonal wind in regions of positive (negative) shear, ...'
It is changed as suggested. [p.8, L217–218]

13) p.7, l.212: February –> March
Thank you! It is corrected. [p.8, L234]

14) p.7, l.195-213: I miss some linking of these results with recent literature in this paragraph:
–> How do these results compare to Lin et al. (2019)? Especially for MRG, IG and CGW, it should be highlighted your results build upon those by Lin et al. (2019) (their Fig. 3 is quite similar to yours), that focus mostly on MRG in their discussions. Your eq. wave differentiation is more detailed, which is a big plus.
–> Regarding your Kelvin wave forcing results, I find it really neat that it helps maintain the two westerly jets. This was suggested/hinted by Li et al. (2020), who studied the longlasting westerly jet around 20hPa, but your results really confirm this is the case. Li et al. (2020) also showed above-average upper-tropospheric Kelvin wave activity linked to El Nino. This should be mentioned/discussed within this paragraph.

–> Also, this is a very long paragraph, could be easily split into 2-3.

Thank you for your detailed comments. We modified the paragraph as suggested:

- We included a comparison between Fig. 2 and Lin et al. (2019)'s Fig. 3. We also emphasized that the contribution to the QBO disruption from the IG waves is first shown in this study.
- We included a discussion on the long-lasting jet near 20 hPa and its contribution from Kelvin waves, referring to Li et al. (2020).
- We split a paragraph into three.

[p. 8, L221–235]

15) Fig. 3: I suggest to move the climatology plot to the supplement (as it doesn't have crucial information for the main text), and make the rest bigger. Right now this figure is too busy and it's easy to get lost within, specially with the duplicate letters for the sub-panels.

Thank you for your comment. Although we respect the reviewer's suggestion, we decided to keep the figure in the revised manuscript, as two figures (disruption and climatology) together can clearly show how the structures of the wind and wave forcings are unusual during the disruption. We instead changed letters for sub-panels and adjusted each figure to increase visibility as much as possible. [Fig. 3]

16) p.9, l.253-254: Mention to Lin et al. (2019) needed here. Your results are in line with theirs, plus the addition of IG in the preconditioning.

We mentioned Lin et al. (2019) in the revised manuscript as suggested. [p. 10, L279–280]

17) p.10, l.294: origination –> just 'origin'

It is corrected. [p.11, L325]

18) p.11, l.324: I suggest making this (dividing by density) for all figures: the vertical/ horizontal arrows without scaling for pressure in the previous plots, may lead the reader to underestimate upper level wave propagation at first glance, specially if one is quickly comparing upper tropospheric and stratospheric levels without paying much attention to

the legend.

However, this is not a must: the figures have the divergence in pressure-independent units, and the units of the arrows are clearly shown - more than enough to correctly interpret everything with basic knowledge about EP flux.

Thank you for your comment. Following the reviewer's suggestion, we made all figures divided by air density except for Fig. 2, because the EPF for the CGWs in Fig. 2 divided by air density was too strong in the upper stratosphere. [Figs. 5, 7, and 11]

19) p.12, l.369-374: This belongs in the methods section

We moved the statements to the methods section (Sect. 2.5). [p. 7, L190–198]

20) Fig. 8(c-d): I suggest to, apart from making the c-d plot smaller, put the timeseries next to each other, or even merge them into one continuous timeseries.

Thank you for your suggestion. We merged Figs. 8c and 8d into one continuous timeseries. [Fig. 8]

21) p.13, l.379-380: You may easily add a climatology line into Fig. 8(c-d)

Following the reviewer's suggestion, we added a climatology line with ±1 standard deviation range in Fig. 8c. [Fig. 8]

22) p.13, l.381: Could the authors perhaps provide a daily timeseries of MRG EPD (in the boxed region) to compare with the q timeseries to assess this as a source? Would be a nice addition as Fig. 8(e-f)

Thank you for the good suggestion. Following the reviewer's suggestion, we added the daily EPD time series for the MRG waves as a new figure Fig. 8d, and included a discussion that the magnitude of the positive EPD increases as the number of grids with $\bar{q}_\phi < 0$ increases. [Fig. 8; p. 13, L406–407]

23) Fig. 9: I suggest to translate the y-axis unit (mean rain rates) into something more relatable, e.g. mm/day

Thank you for the good point. The unit is changed to "mm day$^{-1}$" in the revised manuscript. [Fig. 9]

 This was linked by Li et al. (2020) to the El Nino event that winter. Perhaps it would be useful to add a discussion somewhere in this section, about the overall increase in eq. wave activity and precipitation together with El Nino.

Also a mention to Barton and McCormack (2017) could be added.

Thank you for your suggestion! We included a discussion on the enhanced Kelvin wave activity related to the El Niño event and the overall enhancement in the equatorial wave activity by referring to Li et al. (2020). We did not include a paper by Barton and McCormack (2017) here because as far as we know, their focus was not on the equatorial waves but on the midlatitude Rossby waves and their equatorward propagation. [p.15, L447–449]

25) p.14, l.440-442: There is a lot of new information and little justification about the source level here, I suggest you detail more about this in the methods section, and refer to the corresponding methods section when you start with this figure.

Following the reviewer's suggestion, we included the details about the source level in the method section (Sect. 2.3) by including a new figure (Figure S1) and refer to them when we started to describe Fig. 12 in the revised manuscript. The changes in the direction of the vertical EPF and the sign of the EPD appear at 150 hPa and below (140 hPa and above) for $H_e$ ($H_w$) waves. Our focus is on the westward waves, so we simply assumed the source level of $H_e + H_w$ as 140 hPa. [p.6, L160–162; p.15, L471]

26) p.15, l.446: A non-expert will need more explanation about the source level to understand the attribution made in this sentence. Again, this could be already detailed in the methods section and referred to here.

As discussed in the Comment #25, we included more explanation in the methods section. [p.6, L160–162; p.16, L475]

27) Figure 13 and p.15, l. 447-460: I don't doubt the validity of your results and conclusions regarding Fig. 13, but it would be much easier to interpret if you showed the same plots for He and Hw separately. Mixing both makes this figure a bit confusing. Separating He and Hw will allow the reader to identify which fraction deviates more from climatology (and gets filtered by the wind shear) in a more straightforward way.

We plotted Fig. 13 mimicking the phase-speed spectrum of the GW parameterization (e.g., Fig. 1 of Beres et al. 2004; Fig. 2 of Kang et al., 2017) which, together with wind profiles, can well

represent the sign and magnitude of the potential GWD. Therefore, we would like to keep Fig. 13 as the original. We understand that some readers might be confused and there could be an offset between eastward and westward waves, so we provided the same figure as Fig. 13 but for the eastward and westward propagating parts separately, in the revised supplement (Figure S9). Please note that the figure is not separated into $H_e$ and $H_w$ but into the eastward and westward waves. This is because we defined $H_e$ where $k > 0$ and $H_w$ where $k < 0$ based on the assumption that the zonal wind is close to zero near the equator, but precisely, the direction of the waves is determined by the wind speed at the altitude where the waves are generated (i.e., source level). Since we assumed the source level as 140 hPa (please refer to Response #25), the spectrum where the phase speed is greater (smaller) than zonal wind at 140 hPa is defined as eastward (westward) waves. [p.16, L492–493; Fig. S9]

28) p.15, l.462-467: You discuss the Kawatani paper in the next paragraph. Barton and McCormack (2017) showed important ENSO influence on the background winds and momentum fluxes below 30hPa (see e.g. their plots 3 and 4). It would be worth to add it into your discussion.

Following the reviewer's suggestion, we mentioned ENSO influence on the background winds in the revised manuscript referring to Barton and McCormack (2017). However, ENSO influence on the momentum fluxes below 30 hPa in Fig. 4 of Barton and McCormack (2017) is shown for the horizontal component only, while IG waves have dominant vertical component. Therefore, no further discussions regarding to the work by Barton and McCormack (2017) are made. [p.16, L502–504]

29) p.16, l.488-490: Please detail a bit more what convective source and WFRF mean for the non-expert. Convective source spectrum is related to the movement of convection itself, WFRF to the GWs emitted from it, correct?

Yes, it is correct. The convective source spectrum is about the size, magnitude, and moving speed of the convection and the WFRF is about the shape of the GW spectra emitted from the convection, which is related to the vertical configuration of the convective heating, the critical-level filtering, and resonance between the forced mode and natural mode. The combined effect of the two determines the magnitude and spectral shape of the CGW momentum flux at the source level. We included more explanation in the revised manuscript, as suggested. [p.17, L522–526]

30) p.16, l.496: higher static stability at which height range? Intuitively, deeper convection is related to tropospheric instability.

Thank you for your comment! We found that the original statement was somewhat unclear. We included the height range, 200–300 hPa, and modified the sentence more clearly referring to He et al. (2019) in the revised manuscript. According to He et al. (2019), high tropical static stability appears under global warming because of the enhanced surface temperature resulting in a smaller moist adiabatic temperature lapse rate. Please note that 2016 is the warmest year on record for the global-mean surface temperature (GISTEMP Team, 2020). [p.17, L531–533]

31) p.16, l.497: please use relative terms: warmer / colder

Thank you for your suggestion, but the related sentence was removed during the revision process.

32) p.16, l.497-499: Rephrase this sentence to make it clear that El Nino increases overall amount of convection in the tropics. There must be earlier studies (probably mentioned in the Domeisen review paper) providing this relation to say this with more certainty than 'possibly triggered by El Nino'

Thank you for your suggestion. We rephrased the sentence that El Niño increases the overall amount of convection in the tropics referring to Geller et al. (2016) and Kawatani et al. (2019). [p.17, L529–531]

**References**

Beres, J. J., M. J. Alexander, and J. R. Holton: A method of specifying the gravity wave spectrum above convection based on latent heating properties and background wind. J. Atmos. Sci., 61, 324–337, doi:10.1175/1520-0469(2004)061<0324:AMOSTG>2.0.CO;2, 2004.

Geller, M. A., T. Zhou, and W. Yuan: The QBO, gravity waves forced by tropical convection, and ENSO, J. Geophys. Res. Atmos., 121, 8886–8895, doi:10.1002/2015JD024125, 2016.

GISTEMP Team: GISS Surface Temperature Analysis (GISTEMP), version 4. NASA Goddard Institute for Space Studies. Dataset accessed 2020-09-10 at https://data.giss.nasa.gov/gistemp/, 2020.

He, C., Y. Wang, and T. Li: Weakened impact of the developing El Niño on tropical Indian Ocean climate variability under global warming, J. Climate, 32, 7265–7279, https://doi.org/10.1175/JCLI-D-19-0165.1, 2019.

**Response to Reviewer #2's Comments**

**General Comment:**

This paper gives a comprehensive overview of the different wave forcings that act during the 2016 QBO disruption. The paper discusses the effect of wave forcing by extratropical Rossby waves, equatorial waves, and small scale convective gravity waves, and is therefore the most complete description of the QBO disruption so far. The paper is very well written and fits well into the scope of ACP. The paper is recommended for publication after addressing my minor comments as detailed below.

**Main Comments:**

1) For the spectral analysis time segments of 90 days are used after applying sine and cosine windows at the first and last 30 days. According to Parseval's theorem, this will lead to an underestimation of spectral amplitudes, and, on average, EPF and EPD will be ~30% underestimated. This relatively small effect will not affect the basic results of the paper, but it should be mentioned.

Thank you for pointing out this part. We already recognized that the variance of the 90-day time series becomes two-thirds of the original variance after applying sine and cosine windows at the first and last 30 days, respectively. Therefore, in our calculation, EPF and EPD were multiplied by a scale factor of 3/2, which was not mentioned clearly in the original manuscript. Although no specific sentences on the scaling were given in the paper of Kim and Chun (2015), it was mentioned in a recent paper by Kim et al. (2019). The calculation of the EPF and EPD is clarified in the revised manuscript referring to Kim et al. (2019). [p.5, L148]

2) I am not sure whether the preconditioning of the QBO disruption by MRG EPD and IG EPD in October 2015 is a reliable result. Firstly, the magnitude of EPD is quite small, secondly, the EPD in October 2015 looks different in ERA-Interim, as can be seen in the supplement.

Thank you for asking an important question.

- With regard to your first question, the sum of the MRG and IG wave forcings are found to be ~-0.9 (-1.2) m s$^{-1}$ mon$^{-1}$, which is ~61% (55%) of the total negative forcing in October

(November) 2015. We think that these magnitudes are considerably larger than what we expect and cannot be negligible.

- As for your second question, we included the time series of zonal wind, zonal wind tendency, and each wave forcing using ERA-I data as a new figure in the supplement (Figure S6). The contribution of each wave forcing to the total negative wave forcing is given in the below table (Table A1). It is found that the sum of MRG wave forcing and IG wave forcing in October (November) 2015 using ERA-I data is $\sim$-1.0 (-1.3) m s$^{-1}$ mon$^{-1}$, 61% (53%) of the total negative forcing, which is very similar to that using MERRA-2 data, although MRG (IG) wave forcing is somewhat greater (smaller).

Based on the large contribution by MRG and IG waves and its consistency between datasets, we concluded that MRG and IG waves precondition the zonal wind in the early stage of the QBO disruption. Please also refer to the response to the Specific Comment #6. [Fig. S6]

**Table A1.** The same as Table 1 but using ERA-I data except for the climatology.

| 2015–16 | Oct 2015 | Nov 2015 | Dec 2015 | Jan 2016 | Feb 2016 | Mar 2016 |
|---|---|---|---|---|---|---|
| **MRG** | -0.7 (41%) | -0.8 (32%) | -0.8 (24%) | -0.9 (18%) | -1.5 (27%) | -1.1 (28%) |
| **IG** | -0.3 (19%) | -0.5 (21%) | -0.3 (9%) | -0.1 (2%) | -0.3 (6%) | -0.6 (16%) |
| **Rossby** | -0.6 (40%) | -1.2 (47%) | -2.2 (67%) | -4.0 (80%) | -3.7 (67%) | -2.2 (56%) |
| **Kelvin** | 1.1 | 0.7 | 1.0 | 1.6 | 1.2 | 0.9 |
| **Rossby-Y** | -0.5 (34%) | -1.0 (38%) | -1.9 (58%) | -3.4 (70%) | -3.2 (58%) | -1.9 (48%) |
| **Rossby-Z** | -0.1 (6%) | -0.2 (9%) | -0.3 (9%) | -0.5 (10%) | -0.5 (9%) | -0.3 (8%) |

**Specific Comments:**

1) p2, l33: You may want to include more recent work on the effect of the QBO on surface weather and climate, for example Kidston et al. (2015), or Gray et al. (2018).

   Kidston, J., A. A. Scaife, S. C. Hardiman, D. M. Mitchell, N. Butchart, M. P. Baldwin, and L. J. Gray (2015), Stratospheric influence on tropospheric jet streams, storm tracks and surface weather, Nat. Geosci., 8, 433-440.

   Gray, L. J., J. A. Anstey, Y. Kawatani, H. Lu, S. Osprey, and V. Schenzinger (2018), Surface impacts of the Quasi Biennial Oscillation, Atmos. Chem. Phys., 18, 8227-8247.

Thank you for suggesting more recent works. Those are included in the revised manuscript.

[p.2, L33]

2) p3, l73: The reference Evan et al., JAS, 2012 does not fit here. Evan et al., JAS, 2012 discuss intermediate-scale tropical inertia-gravity waves of horizontal wavelengths in the 5000km range, and not the effect of small scale gravity waves.

This reference should be replaced by Evan et al., JGR, 2012 which is a WRF simulation of the QBO forcing by small scale gravity waves of horizontal wavelengths >270km. In addition to the listed model studies you should also include observational evidence of the effect of small scale gravity waves, for example Ern et al., JGR, 2014.

Evan, S., M. J. Alexander, and J. Dudhia (2012), WRF simulations of convectively generated gravity waves in opposite QBO phases, J. Geophys. Res., 117, D12117, doi:10.1029/2011JD017302.

Ern, M., F. Ploeger, P. Preusse, J. C. Gille, L. J. Gray, S. Kalisch, M. G. Mlynczak, J. M. Russell III, and M. Riese (2014), Interaction of gravity waves with the QBO: A satellite perspective, J. Geophys. Res. Atmos., 119, 2329-2355, doi:10.1002/2013JD020731.

Thank you for pointing out the mistake and suggesting appropriate papers. The reference Evan et al. (JAS, 2012) is changed into Evan et al. (JGR, 2012). We also included observational evidence by Ern et al. (2014) as suggested. [p.3, L73]

3) p3, l88: "All" gravity waves is not correct! As stated in l.82/83, this paper discusses only small scale gravity waves of horizontal wavelengths <100-200km. However, there is also considerable QBO forcing by small scale gravity waves of horizontal wavelengths >200km, as can be seen from Evan et al., JGR, 2012.

We agree with you, but the related sentence was removed during the revision process.

4) p4, l106: On the selection of WQBO cases: Please state more clearly that the definition of the WQBO as used in this paper focuses on QBO situations that are comparable to that of the 2016 QBO disruption. Generally, there should be WQBO periods or WQBO onsets also in other months.

Thank you for your comment. We clarified that there should be WQBO phases in other seasons, but we only focus on NH winter to compare with the QBO disruption case. [p.4, L111–112]

5) p5, l134/135: As EPF is obtained by summing in the spectral space (l.142), applying cosine windows will lead to an underestimation of spectral amplitudes, and also of EPF, and EPD. See Main Comment (1).

As discussed in the Main Comment #1, we included the sentence that EPF is multiplied by a scale factor of 3/2 to conserve its original variance. [p.5, L148]

6) p9, l253/254: I am not sure whether it is a reliable result that IG and MRG would act as a preconditioning at 40hPa in October 2015 before Rossby waves can take effect! Please note that Fig.4b (for MERRA2) and Fig.S5 in the supplement (for ERA-Interim) look quite different! The preconditioning effect that you suggest seems to be much weaker than the differences between the two reanalyses.

Thank you for your comment. Please note that the time series of Fig. S5 in the original supplement (Fig. S7 in the revised supplement) is not for ERA-I but for MERRA-2 averaged over 5°S–10°S. As discussed in the Main Comment #2, the sum of MRG wave forcing and IG wave forcing is quite similar between MERRA-2 and ERA-I data in October–November 2015. The wave forcing by IG waves using ERA-I is somewhat smaller than that using MERRA-2, possibly due to a coarser horizontal resolution (0.75°). The related sentence is included in the revised manuscript. We also clarified in the figure caption that Fig. S5 (Fig. S7 in the revised manuscript) is for MERRA-2 data. [p.10, L306–307; Figs. S6–S7]

7) p10, l281-285: Another reason for this difference could be the part of the gravity wave spectrum that is neither covered by the CGW scheme, nor resolved or parameterized in MERRA2. It should be emphasized that MERRA2 is not a free-running model! There will be model imbalances that are caused by data assimilation. Data assimilation can therefore correct misrepresentations of the gravity wave forcing by the nonoragpic GWD parameterization.

We agree with you that data assimilation corrects misrepresentations of gravity wave forcing by GWD parameterization which could stem from the missing GW spectrum in the parameterization. However, this fact may not be strongly related to the difference between the parameterized GWD from MERRA2 and CGWD from the offline parameterization, because the analysis increment is not included in the parameterized GWD but is provided separately as a different variable named "total eastward wind analysis tendency" in MERRA-2 (GMAO, 2015).

8) Fig.6: Here you identify source regions of Rossby waves by positive EPD coinciding with upward directed EPF. At 15N around 150hPa EPF is directed downward coinciding with negative EPD. Do you think this is another source region of Rossby waves? As EPF is directed downward, it looks like these waves cannot propagate into the stratosphere.

We agree with you, and the related sentence is deleted in the revised manuscript.

9) p11, l327: that the positive EPD region is a source region of the westward-propagating waves. -> that the positive EPD region should be a source region of westward-and upward propagating waves.

Thank you for your suggestion. It is modified. [p.12, L358]

10) p12, l370: It should be mentioned that Eq.(3) includes both barotropic and baroclinic instability. Did you check which term is stronger - the barotropic term (meridional gradients), or the baroclinic term (vertical gradients)? Coy et al., 2017 claimed that barotropic instability would be stronger. In your Fig.8, this does not fit the title and the figure caption saying "baroclinic instability".

Thank you for your suggestion. In the original manuscript, it was mentioned that the $\bar{q}_\phi$ term is dominated by the barotropic instability; but it was not highlighted. Figure A1 below shows the same as Figs. 8a–b but excluding the third term in the righthand side of Eq. (3), which is very similar to Figs. 8a–b. This implies that the barotropic term is much larger than the baroclinic term. This result is consistent with the finding of Garcia and Richter (2019), who showed that the reversal of the vorticity gradient was almost completely dominated by the behavior of the barotropic part, specifically by the behavior of the meridional curvature of the zonal wind. We moved explanation of Eq. (3) to the methods section (Sect. 2.5 in the revised manuscript) and stated clearly that the Eq. (3) is dominated by the barotropic instability. Also, "baroclinic instability" in Fig. 8 is changed into "barotropic instability". [p.7, L198–199; Fig. 8]

**Figure A1.** The same as Figs. 8a–b but for $\bar{q}_\phi$ excluding the third term in the righthand side of Eq. (3).

11) p17, l513: barotropic instability -> barotropic and/or baroclinic instability

As discussed in the Specific Comment #11, we stated clearly in the revised manuscript that $\bar{q}_\phi$ term is dominated by the barotropic instability. Therefore, we would like to keep "barotropic instability" here.

**Technical comments:**

1) p5, l124: the parameter \hat{f} is not used in Eq.(1)

Thank you for pointing out this error! "Eq. (1)" is changed to "$F^\phi$ and $F^z$" in the revised manuscript. [p.5, L128]

2) p6, l175: is lower than 700 hPa -> is at altitudes lower than 700 hPa

Thank you! It is corrected. [p.6, L182]

3) Fig.1: colorbar should be m/s, but is m/s/month

Thank you! It is corrected. [Fig. 1]

4) Fig.3: one of the colorbars should refer to the wind in m/s, but both colorbars give

Thank you! It is corrected. [Fig. 3]

Thank you! It is corrected. [p.8, L244]

6)   p9, l256/257: Figure S4 is the same figure with Fig. 3 but using ERA-I data. -> Figure S4 shows the same as Fig. 3 but using ERA-I data.

It is modified as suggested. [p.10, L283]

Thank you! It is corrected. [p.10, L291]

Thank you! It is corrected. [Fig. 7]

9)   caption of Fig.7, l799: where the EPD is smaller than -> where the EPD is stronger than

Thank you for pointing out the ambiguous expression. It is changed to "where the EPD is algebraically smaller (more negative) than the climatology", not only in the caption of Fig. 7 but also in that of Figs. 5 and 11 for consistency. [Figs. 5, 7, 11]

Thank you! It is corrected. [p.14, L417]

Thank you! It is corrected. [Fig. 9]

**References**

Global Modeling and Assimilation Office (GMAO): MERRA-2 tavg3_3d_udt_Np: 3d,3-Hourly,Time-Averaged,Pressure-Level,Assimilation,Wind Tendencies V5.12.4, Greenbelt, MD, USA, Goddard Earth Sciences Data and Information Services Center (GES DISC), accessed 28 September 2020, 10.5067/CWV0G3PPPWFW, 2015.

Kim, Y.-H., Kiladis, G.N., Albers, J.R., Dias, J., Fujiwara, M., Anstey, J.A., Song, I.-S., Wright,

C.J., Kawatani, Y., Lott, F. and Yoo, C.: Comparison of equatorial wave activity in the tropical tropopause layer and stratosphere represented in reanalyses. Atmos. Chem. Phys., 19, 10027–10050, 2019.

**Response to Reviewer #3's Comments**

**General Comment:**

Summary: This paper provides a detailed examination of the equatorial wave structures and their evolution during October 2015 through February 2016, a time when the quasi-biennial oscillation experienced a significant disruption. This investigation, based on MERRA-2, a global assimilation system, breaks down the wind, temperature, and precipitation fields into Rossby, Mixed Rossby Gravity (MRG), inertia-Gravity (IG), and Convective Gravity (CG) waves. A novel aspect of this work is the use of a convective gravity wave parameterization to calculate the CG wave effects. The different Eliassen Palm (EP) fluxes and their divergences are evaluated. Quantities calculated for the QBO disruption are compared to their corresponding climate signatures. The results show how during October-November of 2015 MRG and IG waves acted to precondition the QBO winds before the strong Rossby waves that occurred in 2016 and created the anomalous QBO easterlies. Why these waves were stronger than usual during this time is still unknown.

Strengths: This work provides a comprehensive view of the UT/LS equatorial waves during the QBO disruption. It expands on the work of Lin et al. (2019) by including aspects of the tropospheric forcing by precipitation and the addition of CG wave model and also differs in the choice of assimilation system from Lin et al. (2019). Figure 16 provides an especially useful summary of the changes roles played by the different waves.

Weaknesses: No major weaknesses. There are a few points that could be improved for clarity that are detailed below. There are a large number of figures. These provide a comprehensive record but it can be difficult for readers to locate the features of interest as described in the text.

Recommendation: Publish after minor revisions noted below. This is a well written, well organized, manuscript. The figures are appropriately captioned and the abstract accurately summaries the work. The topic should be of interest to many readers of ACP interested the the QBO and equatorial waves.

**Comments:**

1) Line 44: "El Nino" Coy et al. (2017) only mentioned a possibility of an ENSO connect. The terms El Nino or ENSO are not found in Osprey et al. (2016) so this sentence should be rewritten.

Thank you for pointing this out! The reference by Osprey et al. (2016) is deleted in the revised manuscript.

2) Line 58: Should be "Coy et al. (2017)". There is a Coy et al. (2016) describing the MERRA-2 QBO before the disruption that could probably be mentioned somewhere in the data section: Coy, L., K. Wargan, A. M. Molod, W. R. McCarty, and S. Pawson. 2016. "Structure and Dynamics of the Quasi-Biennial Oscillation in MERRA-2." J. Climate, 29:14: 5339-5354 [10.1175/jcli-d-15-0809.1]

Thank you for pointing out this error! It is corrected. [p.2, L58]

3) Line 67: Each MERRA-2 data set has a DOI number that researchers are encouraged to reference and clarifies exactly what data set was used. For example from https://disc.gsfc.nasa.gov/datasets/M2I3NVASM_5.12.4/summary?keywords=%22MERRA-2%22 To cite the data in publications: Global Modeling and Assimilation Office (GMAO) (2015), MERRA-2 inst3_3d_asm_Nv: 3d,3-Hourly,Instantaneous,Model-Level,Assimilation,Assimilated Meteorological Fields V5.12.4, Greenbelt, MD, USA, Goddard Earth Sciences Data and Information Services Center (GES DISC), Accessed: [Data Access Date], 10.5067/WWQSXQ8IVFW8

Thank you for your important point. We added MERRA-2 data reference in the revised manuscript. [p.3, L68]

4) Lines 248-249: How is the budget formulated so that resolved MRG and IG wave forcing acts to "enhance" the momentum budget residual, REQ? Maybe this could be rewritten for clarity.

Thank you for your good comment. We found that the original sentence was unclear. We intended to say that the strong MRG and IG wave forcing may account for the relatively strong negative REQ in November 2015. We changed the sentence as follows: "IG waves exert a strong negative forcing in November 2015, which might be related to the enhancement of

negative REQ at 40 hPa along with the MRG wave forcing". [p.9, L272–273]

5) Lines 277-287: This is a good discussion of the CGWD. It would help to see the vertical zonal mean zonal wind shear at 40 hPa as a part of Fig. 4 as that should determined in large part the GWD forcing. In addition the meridional shear of the zonal mean zonal wind across the equator might correspond to the changes in the Rossby wave forcing and be helpful to see plotted.

Thank you for your good suggestion. During the revision process, we plotted meridional wind shear across the equator and vertical wind shear averaged over 5°N–5°S, which are included as a new figure (Fig. 4d) in the revised manuscript. We found that the temporal evolution of CGW (Rossby wave) forcing is similar to that of vertical (meridional) wind shear. The related discussion is included in the revised manuscript. [p.10, L303–306]

6) Lines 357-362: Figures 3, 4, and 7 all illustrate aspects of MRG waves, however it is difficult to put together a consistent picture of support for the mid-jet easterly acceleration described here. Most of the easterly acceleration appears to take place in the regions of strong wind shear, not mid-jet. In particular the contribution from MRG waves in Fig. 4 at 40 hPa is small and appears nearly constant in time. The mid-jet should be identified more quantitatively and MRG wave aspects calculated with respect to the jet at each month, at least for the earlier months, to justify this conclusion.

Thank you for your comment. Although the MRG wave forcing is generally strong in the regions of strong wind shear, the easterly acceleration by the MRG waves in the mid-jet (5°N–5°S) is evident at 40 hPa in October, November, and December 2015 with magnitudes of -0.4, -0.6, and -0.6 m s$^{-1}$ mon$^{-1}$, respectively (Table 1). The magnitude of mid-jet in 5°N–5°S is 12.3, 11.1, and 8.2 m s$^{-1}$ in October, November, and December, respectively (Figure 4a), which implies that the MRG wave forcing can at least partly account for the weakening of the mid-jet. During the revision process, we additionally plotted a time–latitude cross section of the MRG wave forcing at the altitude range of 40–60 hPa, where the MRG wave forcing was strong (Figure A2), which also shows a deceleration of the jet core by the MRG waves in the early period. The related discussion is included in the revised manuscript. [p.13, L389–390]

[Figure]

**Figure A2.** Time–latitude cross section of EPD for the MRG waves at 40–60 hPa, superimposed with the zonal-mean zonal wind (contour lines). Positive (negative) zonal winds are plotted with solid (dashed) lines with a contour interval of 2 m s$^{-1}$, and thick contour lines denote a zero-zonal wind speed.

7) Lines 375-392: This instability analysis is based on zonal mean winds. A stronger case for instability might be possible with non-zonally average winds, especially when the focus is on the relatively small region defined by the box in Figure 8.

Thank you for your comment. We found that the number of grids with $\bar{q}_\phi < 0$ is much larger than the original calculation when calculating baroclinic instability based on each longitudinal grid (Figure A3). However, we thought that the background flow defined to extract the MRG waves should be used to evaluate baroclinic instability as a possible source of the MRG waves. Therefore, we would like to keep the calculation of $\bar{q}_\phi$ based on zonal averages.

[Figure]

**Figure A3.** The number of grids where daily-mean $\bar{q}_\phi$ (s$^{-1}$) in each longitude is negative in the boxed region (5°–15°S, 60–90 hPa) normalized by the number of longitudes (nx = 576) in October–November 2015 (red) and its climatology with ±1 standard deviation (gray shading).

8) Lines 443-485: The "...apparent positive wind shear..." Is difficult to find in Fig. 14a.The specific levels should be specified in the text.

Thank you for your comment. The specific levels (140–200 hPa) are included in the revised manuscript. [p.17, L515–516]

9) Lines 499-502: The white and gray curves described in the text appear to be different from the Fig. 15 caption description.

Thank you. The description on the gray curves is modified in the revised manuscript. [p.17, L534]

**Minor Comments:**

1) Line 375: The units in Fig. 8 suggest that $q\_y$ is plotted not $q\_phi$. This is a small point. Perhaps the units could be described in the figure caption.

Thank you for pointing out the error! The unit is corrected and described in the figure caption. [Fig. 8]

2) Lines 732-733: The year is missing from the reference.

Thank you, but we think the year '2011' was included in the original manuscript.

**Response to Reviewer #4's Comments**

**General Comment:**

The authors investigated the relative contribution of each resolved wave and parameterized waves to the QBO disruption in 2015-2016. They have shown that MRG and westward IG weakened the QBO and then led to extratropical Rossby breaking at the QBO jet core at 40 hPa. They also investigated the roles of CGWs obtained from an offline CGW parameterization that author's group has developed and showed the importance of variable wave sources. There have been several studies to investigate the mechanism of the 2015-2016 QBO disruption. I think this paper is the most comprehensive study among them. I believe this paper is suitable for the publication in ACP. My recommendation is published after very minor revision. I have a few comments added below.

**Comments:**

1) MRG are confined to the range |k|<=20 and omega <0.75 cpd in the symmetric spectrum. I think zonal |k|<=20 is a little wide for the MRG. Presumably |k|<=~10 would be better. Westward IGWs should be included in this definition. How much do the results depend on the ranges of |k|? I guess the relative contribution of MRG, shown in Table 1 and Figure 4, would be changed. One good point to answer this concern is to mention the dominant zonal wavenumber ranges for the MRG to force the QBO. I guess 3 < k < 6, but am not sure. I would suggest authors at least to mention the dependence of |k| selection to the quantitative results.

Thank you for your comment! The MRG waves are confined to the spectral range where the signs of $F^{z1}$ and $F^{z2}$ are the opposite within $|k| \leq 20$ and $0.1 \leq \omega \leq 0.5$ cpd in the anti-symmetric spectrum (Kim and Chun 2015, JGR). Here, the $F^{z1}$ and $F^{z2}$ represent the first and second terms of the vertical EPF ($F^z$). This range is basically within $|k| \leq 10$ as seen in the example of the MRG wave spectrum in Figure A4, which is similar to what the reviewer expected. During the revision process, we performed a sensitivity test on the EPD for the MRG waves by changing the spectral boundary of the MRG waves as $|k| \leq 10$ and $0.1 \leq \omega \leq 0.5$ cpd in the anti-symmetric spectrum. It is found that the EPDs of the MRG waves in October

and November 2015 are -0.437 and -0.5883 m s$^{-1}$ mon$^{-1}$ (originally: -0.433 and -0.5876 m s$^{-1}$ mon$^{-1}$), respectively, implying that the difference is within 1%. The dominant zonal wavenumber ranges for MRG waves are mentioned in the revised manuscript. [p.5, L144–145]

[Figure]

**Figure A4.** Spectral density of the EPD for the MRG waves averaged over 4ºN–4ºS at 40–60 hPa in November 2015.

2) L216: "The required wave forcing term (REQ) is calculated as a residual by subtracting the advection terms from the zonal wind tendency in the TEM equation"
   When calculating REQ, do the authors consider the first term on the left of Eq. (1), that is meridional advection term, which is normally very small near the equator? In my experiences, the meridional advection term has also some values off the equator even at ~5 degrees, which cannot be sometimes negligible.

Yes, we also consider the meridional advection term, so the REQ is calculated as a residual by subtracting both the meridional and vertical advection terms from the zonal wind tendency. It is clarified in the revised manuscript. [p.8, L239]

3) L258: "although the magnitudes of the REQ and wave forcing (vertical advection) in ERA-I is generally stronger (weaker) than that in MERRA-2" I guess one possible reason for this is the different values of w* between MERRA-2 and ERA-I. As you know, the representation of BDC is quite different quantitatively among reanalyses as the S-RIP

project has indicated.

We agree with you. As the reviewer mentioned, $\overline{w}^*$ value in the reanalysis is well known for its large spread. In addition to the differences in $\overline{w}^*$, a large spread in the vertical wind shear, which is generally related to the vertical resolution of the data, is also a possible reason for the discrepancy in the vertical advection term, which is supported by Figure 5 of Kim and Chun (2015, ACP). The related discussion is included in the revised manuscript. [p.10, L285–286]

4) I would suggest to refer the paper by Dunkerton (2016, GRL, https://doi.org/10.1002/2016GL070921). Dunkerton's paper, published just after the QBO disruption, discussed some presumable mechanisms, which would be now useful for the current study.

Thank you for suggesting a paper. The paper by Dunkerton (2016) is now included in the revised manuscript. [p.2, L44]

5) Figure 4(c): The explanation lines of Rossby-Y & Rossby-Z are hard to see. Please expand the lines.

Thank you for pointing this out! The explanation lines in Figure 4c are expanded in the revised manuscript. [Fig. 4]

**References**

Kim, Y.-H., and H.-Y. Chun: Momentum forcing of the quasi-biennial oscillation by equatorial waves in recent reanalyses. Atmos. Chem. Phys., 15, 6577–6587, doi:10.5194/acp-15-6577-2015, 2015.

---

## Author Comment (AC4) · 13 Oct 2020

General Comment:

The authors investigated the relative contribution of each resolved wave and parameterized waves to the QBO disruption in 2015-2016. They have shown that MRG and westward IG weakened the QBO and then led to extratropical Rossby breaking at the QBO jet core at 40 hPa. They also investigated the roles of CGWs obtained from an offline CGW parameterization that author's group has developed and showed the importance of variable wave sources. There have been several studies to investigate the mechanism of the 2015-2016 QBO disruption. I think this paper is the most compre-

hensive study among them. I believe this paper is suitable for the publication in ACP. My recommendation is published after very minor revision. I have a few comments added below.

Response: First of all, we would like to thank the reviewer for the time spent on reviewing our long manuscript. All reviews have been beneficial and made us aware of important points which had to be addressed. We, the authors, are therefore thankful for the reviewer's contribution to improve the manuscript. We carefully addressed all comments and tried our best to improve the manuscript based on the suggestions and comments.

Comments:

1. MRG are confined to the range |k|<=20 and omega <0.75 cpd in the symmetric spectrum. I think zonal |k|<=20 is a little wide for the MRG. Presumably |k|<=∼10 would be better. Westward IGWs should be included in this definition. How much do the results depend on the ranges of |k|? I guess the relative contribution of MRG, shown in Table 1 and Figure 4, would be changed. One good point to answer this concern is to mention the dominant zonal wavenumber ranges for the MRG to force the QBO. I guess 3 < k < 6, but am not sure. I would suggest authors at least to mention the dependence of |k| selection to the quantitative results.

Response: Thank you for your comment! The MRG waves are confined to the spectral range where the signs of Fz1 and Fz2 are the opposite within $|k| \le 20$ and $0.1 \le \omega \le 0.5$ cpd in the anti-symmetric spectrum (Kim and Chun 2015, JGR). Here, the Fz1 and Fz2 represent the first and second terms of the vertical EPF (Fz). This range is basically within $|k| \le 10$ as seen in the example of the MRG wave spectrum in Figure A4, which is similar to what the reviewer expected. During the revision process, we performed a sensitivity test on the EPD for the MRG waves by changing the spectral boundary of the MRG waves as $|k| \le 10$ and $0.1 \le \omega \le 0.5$ cpd in the anti-symmetric spectrum. It is found that the EPDs of the MRG waves in October and November 2015 are -0.437 and

-0.5883 m/s/mon (originally: -0.433 and -0.5876 m/s/mon), respectively, implying that the difference is within 1%. The dominant zonal wavenumber ranges for MRG waves are mentioned in the revised manuscript. [p.5, L144–145]

2. L216: "The required wave forcing term (REQ) is calculated as a residual by subtracting the advection terms from the zonal wind tendency in the TEM equation" When calculating REQ, do the authors consider the first term on the left of Eq. (1), that is meridional advection term, which is normally very small near the equator? In my experiences, the meridional advection term has also some values off the equator even at ∼5 degrees, which cannot be sometimes negligible.

Response: Yes, we also consider the meridional advection term, so the REQ is calculated as a residual by subtracting both the meridional and vertical advection terms from the zonal wind tendency. It is clarified in the revised manuscript. [p.8, L239]

3. L258: "although the magnitudes of the REQ and wave forcing (vertical advection) in ERA-I is generally stronger (weaker) than that in MERRA-2" I guess one possible reason for this is the different values of w* between MERRA-2 and ERA-I. As you know, the representation of BDC is quite different quantitatively among reanalyses as the S-RIP project has indicated.

Response: We agree with you. As the reviewer mentioned, w* value in the reanalysis is well known for its large spread. In addition to the differences in w*, a large spread in the vertical wind shear, which is generally related to the vertical resolution of the data, is also a possible reason for the discrepancy in the vertical advection term, which is supported by Figure 5 of Kim and Chun (2015, ACP). The related discussion is included in the revised manuscript. [p.10, L285–286]

4. I would suggest to refer the paper by Dunkerton (2016, GRL, https://doi.org/10.1002/2016GL070921). Dunkerton's paper, published just after the QBO disruption, discussed some presumable mechanisms, which would be now useful for the current study.

Response: Thank you for suggesting a paper. The paper by Dunkerton (2016) is now included in the revised manuscript. [p.2, L44]

5. Figure 4(c): The explanation lines of Rossby-Y & Rossby-Z are hard to see. Please expand the lines.

Response: Thank you for pointing this out! The explanation lines in Figure 4c are expanded in the revised manuscript. [Fig. 4]

References

Kim, Y.-H., and H.-Y. Chun: Momentum forcing of the quasi-biennial oscillation by equatorial waves in recent reanalyses. Atmos. Chem. Phys., 15, 6577–6587, doi:10.5194/acp-15-6577-2015, 2015.

4N-4S, 40-60hPa            EPD

Frequency (cpd)

0.80

0.60

0.40

0.20

0.00

-15  -10  -5   0    5   10   15

Zonal wavenumber

-2   -1  -0.5 -0.2 -0.1 -0.05 -0.02 -0.01 0.01 0.02 0.05 0.1 0.2 0.5  1   2     $(10^{-3}\ m\ s^{-1}\ mon^{-1}\ /\ cyc\ day^{-1})$

**Fig. 1.** Figure A4. Spectral density of the EPD for the MRG waves averaged over 4°N–4°S at 40–60 hPa in November 2015.